


# Scenario-based multi-risk assessment from existing single-hazard vulnerability models. An application to consecutive earthquakes and tsunamis in Lima, Peru

Juan Camilo Gómez Zapata[1, 2], Massimiliano Pittore[1, 3], Nils Brinckmann[4], Juan Lizarazo-Marriaga[5], Sergio Medina[5], Nicola Tarque[6, 7], and Fabrice Cotton[1, 2],

[1]Seismic Hazard and Risk Dynamics, GFZ German Research Centre for Geosciences, Potsdam, 14473, Germany

[2] Institute for Geosciences, University of Potsdam, Karl-Liebknecht-Str. 24-25, Potsdam, 14476, Germany

[3] Institute for Earth Observation, EURAC Research, Viale Druso 1, Bolzano, 39100, Italy

[4] eScience Centre, GFZ German Research Centre for Geosciences, Telegrafenberg, Potsdam, 14473, Germany

[5] Departamento de Ingeniería Civil y Agrícola, Universidad Nacional de Colombia, sede Bogotá, 11001, Colombia

[6] Gerdis Research Group, Civil Eng. Division, Pontificia Universidad Católica del Perú, Av. Universitaria 1801, Lima, Peru

[7] Department of Continues Mechanics and Structures, Universidad Politécnica de Madrid, Calle Aranguren 3, Madrid, Spain

*Correspondence to*: jcgomez@gfz-potsdam.de

**Abstract.** Multi-hazard risk assessments for building portfolios exposed to earthquake shaking followed by a tsunami are usually based on empirical vulnerability models calibrated on post-event surveys of damaged buildings. The applicability of these models cannot easily be extrapolated to other regions of larger/smaller events. Moreover, the quantitative evaluation of the damages related to each of the hazards type (disaggregation) is impossible. To investigate cumulative damage on extended building portfolios, this study proposes an alternative and modular method to probabilistically integrate sets of single-hazard vulnerability models that are being constantly developed and calibrated by experts from various research fields to be used within a multi-risk context. This method is based on the proposal of state-dependent fragility functions for the triggered hazard to account for the pre-existing damage, and the harmonisation of building classes and damage states through their taxonomic characterization, which is transversal to any hazard-dependent vulnerability. This modular assemblage also allows us to separate the economic losses expected for each scenario on building portfolios subjected to cascading hazards. We demonstrate its application by assessing the economic losses expected for the residential building stock of Lima, Peru, a megacity commonly exposed to consecutive earthquake and tsunami scenarios. We show the importance of accounting for damage accumulation on extended building portfolios while observing a dependency between the earthquake magnitude and the losses derived for each hazard scenario. For the commonly exposed residential building stock of Lima exposed to both perils, we find that classical tsunami empirical fragility functions lead to an underestimation of predicted losses for lower magnitudes ($M_w$) and large overestimations for larger Mw events in comparison to our state-dependent models and cumulative damage method.



## 1. Introduction

Cascading natural events, commonly defined as a primary hazard triggering a secondary one, have jointly induced large disasters (Gill and Malamud, 2016). In the case of earthquakes, between 25 and 40% of economic losses and deaths have been reported to result as a consequence of secondary effects, i.e., tsunamis, landslides, liquefaction, fire, and others (Daniell et al., 2017). Well-known examples are the submarine earthquakes and the subsequent tsunamis occurred in 2004 in the India Ocean, in 2011 in Japan, and in 2018 in Palu Bay in Indonesia (Goda et al., 2019). These events not only induced cumulative physical damage on the exposed infrastructure, but also brought drastic socioeconomic cascading effects that are still perceptible today (de Ruiter et al., 2020; Suppasri et al., 2021). Despite the magnitude of such events, multi-hazard risk assessment remains a relatively new research field with still not unified terminologies and approaches (Pescaroli and Alexander, 2018; Tilloy et al., 2019). Nonetheless, a number of studies (e.g., Kappes et al., 2012; Komendantova et al., 2014; Gallina et al., 2016; Julià and Ferreira, 2021; De Angeli et al., 2022; Cremen et al., 2022) have unanimously agreed that more realistic multi-risk evaluations can only be conducted if both (1) multi-hazard (e.g., Marzocchi et al., 2012; Liu et al., 2016) and (2) multi-vulnerability interactions (e.g., Zuccaro et al., 2008; Gehl et al., 2013) are considered altogether. While the former comprises the study of the conditional probabilities of the occurrences of these hazards and their combination, the study of the latter involves reviewing the many classes of vulnerabilities that are associated with an exposed territory.

Therefore, in this study, we narrow down the scope by assuming that a second hazardous event is always triggered after the occurrence of the first one, thus eliminating the need to quantify the probability of this occurring. Thus, we will only focus on the dynamic physical vulnerability and related cumulative damage that a building stock exposed to a close succession of hazardous events might suffer.

In exposure modelling for multi-hazard risk purposes, we can distinguish between two main approaches:

1. Using a single set of building classes, each employing as many fragility/vulnerability models as the natural hazards considered, for example, the HAZUS-MH (FEMA, 2003, 2017); Dabbeek and Silva, (2020); and Dabbeek et al., (2020). They have typically associated sets of fragility functions with equivalent damage states regardless of the hazard. Aligned with this philosophy, the EMS-98 vulnerability classes (Grünthal, 1998) were used by some authors to not only describe the likely damage due to seismic action, but also to classify likely ranges of vulnerabilities to other hazards based on the building's material types (Schwarz et al., 2019; Maiwald and Schwarz, 2019).

2. Jointly applying a number of different building classifications per individual hazard to the same exposed buildings (e.g., Gómez Zapata et al., 2021e; Arrighi et al., 2022). Their associated fragility functions may have different sets of damage states (differing in number and description). Notably, these models are constantly developed and individually validated by experts of each research field.

Although the first type might be useful in the assessment of risk arising from independent hazards, their related sets of fragility models lack multi-hazard calibration and validation and, therefore, do not offer sufficient inputs for assessing the increasing damage from cascading events (Ward et al., 2020).



Moreover, the definition of the damage scale depends on the building type (Hill and Rossetto, 2008) and the likely failure mechanisms that it can experience under the action of specific hazard intensity measures (IM) (Vamvatsikos et al., 2010; Selva, 2013). Therefore, the observable damage features on individual structural or non-structural components that jointly describe a certain damage state can have contrasting descriptions across various hazard-dependent vulnerability types (Gehl and D'Ayala, 2018; Figueiredo et al., 2021) and there is often not a 1:1 relation between them for the case of earthquakes and tsunamis (Bonacho and Oliveira, 2018; Lahcene et al., 2021). The reasons behind such a mismatching between the definitions of damage states may arise from the absence of standard formats for damage data collection across regions and across the several vulnerability types of interest (Mas et al., 2020; Frucht et al., 2021). Notably, the study of Negulescu et al., (2020) found this to be particularly significant for the multi-hazard risk context, stating that the damage states of earthquake and tsunami fragility models can have variable levels of compatibility. This assumption led to contrasting loss estimates with respect the U.S HAZUS approach, which is based on the complete equivalence between damage grades. This background portrays the need to standardise the description of the physical damage through harmonizing scales across several hazard-dependent vulnerabilities, which are inputs for unified methods in multi-hazard risk (Ward et al., 2022).

The earthquake engineering has investigated the cumulative damage expected during seismic sequences (e.g. Papadopoulos and Bazzurro, 2021; Karapetrou et al., 2016; Trevlopoulos et al., 2020), but this concept is rarely considered in other research disciplines. For instance, the physical vulnerability of building portfolios to tsunamis has been typically evaluated through empirical fragility functions derived from post-near-field tsunami surveys. A drawback of these functions is that they have been presented solely as tsunami fragility functions in terms of the inundation depth when in reality these surveys encompassed assets that experienced cumulative damage due to the joint effect of the tsunami-generating earthquake and the tsunami itself (Charvet et al., 2017). Due to this limitation, analytical fragility functions were recently proposed for individual structures (e.g., Attary et al., 2017) and for large-scale building stocks with generalised typologies (Belliazzi et al., 2021). However, as remarked by Attary et al., (2019), using these functions for loss estimation should only be valid for far-field tsunamis, and for near-field events the damage induced by shaking before the tsunami strikes must still be addressed.

To the best of the authors' knowledge, only a few studies have investigated the performance of heterogeneous and large-scale building portfolios for risk estimates subjected to consecutive ground shaking and tsunamis. Hereby, we summarize some of them. In Goda and De Risi, (2018) a rationale was proposed for adopting the larger value of the damage ratios from independent earthquake and tsunami risk computations. In Park et al., (2019) a probabilistic multi-risk approach was presented for a building stock in the USA subjected to spatially uncorrelated seismic ground motions and subsequent tsunamis. This study showed the disaggregation of losses per hazard and per material-based building type across several return periods while assuming statistical independence between their respective damage states. As a common denominator of the aforementioned studies, the cumulative damage and losses from a building portfolio were not assessed. Since these metrics cannot be obtained as the sum of the effects from each individual hazardous event (Bernal et al., 2017; Terzi et al., 2019), it is rather necessary to address the nonlinear damage accumulation on the same exposed assets during the multi-hazard sequences (Merz et al., 2020).


This study contributes to the field by proposing a modular method to to probabilistically integrate sets of single-hazard vulnerability models that are being constantly developed and calibrated by experts from various research fields to be used within a multi-risk context, solve the harmonization problem between existing seismic and tsunami building classification schemes and harmonize the damage state definitions within the two single-hazard vulnerability models. This is done with the purpose of representing the damage distribution resulting after the earthquake (shaking) phase through a damage-updated exposure model whose damage scale is dependent on the classification scheme required for assessing the vulnerability to a triggered (tsunami) event. Complementary, we propose to use state-dependent fragility models that account for the pre-existing damage caused by the first event (shaking). These modules are valuable inputs for ultimately assessing the expected cumulative damage that is expected in consecutive hazard scenarios. We demonstrate the application of this method by investigating the likely cumulative damage on the residential buildings of Lima (Peru) by considering this city's exposure to six mega-thrust earthquake scenarios (main shock) and subsequent tsunamis. Every damage distribution is translated into direct economic losses to gain a comparative risk metric and disaggregate the contribution of each hazard scenario.

## 2. Proposed method

To assess the cumulative damage that is expected to be experienced by a building portfolio during hazardous event sequences, we rely on the principle that its related exposure model is represented by jointly applying existing building classification schemes, one per each individual hazardous scenarios of the cascading sequence. For example, one building that is expected to be affected by a first intensity $IM^A$ (e.g., ground-shaking) and a second one $IM^B$ (e.g., tsunami inundation) is actually classified under two exposure classification schemes ($T_k^A$ and $T_j^B$), respectively, which have attached their related vulnerability modes (Figure 1a). Each scheme contains a set of mutually exclusive, collectively exhaustive building classes $k = \{k_1, \ldots, k_n\}$ and $j = \{j_1, \ldots, j_n\}$ correspondingly.

To assess the expected damage state after the first hazardous event (e.g., ground-shaking), we apply their fragility function $\sum_z p(D_{kz}^A | IM^A)$ which give us the probability that a building $k$, typically assumed to be in an undamaged state 0, ($D_{k0}^A$), changes to a progressive state $z$ due to a hazard intensity $IM^A$ (green part in Figure 1b). For risk assessment, this is completed by the consequence model ($L|D_{kz}^A$), which assigns a loss ratio $L$ of the total replacement cost of class $k$ given the occurrence of a damage state. Thus, the expected loss given a hazard intensity $IM^A$ is calculated considering the contributions from all possible damage states and their probabilities, as per Eq. 1.

$$p(L|IM^A) = \sum_z p(D_{kz}^A | IM^A) p(L|D_{kz}^A) \qquad \text{Eq. 1}$$

If this damaged building portfolio is subjected to the action of a second scenario with a hazard intensity $IM^B$, it would experience cumulative damage moving from a damage state $z$, ($D_{kz}^A$) to a damage state $w$ (but in the domain of the second vulnerability scheme: $D_{jw}^B$). Due to this differential scheme classification, their respective set of damage states may not have




trivial equivalences because they can also have different observable damage features. Therefore, we propose integrating a set of modular components, namely: **(1)** inter-scheme compatibilities between each hazard-dependent exposure classification scheme $p\left(T_k^A \mid T_j^B\right)$ (i.e., purple part in Figure 1b, method originally proposed in Gómez Zapata et al., (2022b), and summarised in Sect. 2.1). Moreover, since the fragility models attached to such schemes may have different numbers of damage states and descriptions, we also propose to obtain **(2)** their related compatibility levels between inter-scheme damage states $p\left(D_{kz}^A \mid D_{jy}^B\right)$

(i.e., red part in Figure 1b, explained in Sect. 2.2). Through these conversions, the damaged updated exposure model resulting from the action of $IM^A$ can be represented in the domain of the reference scheme attached to the second vulnerability to be analysed. Complementary, **(3)** single-hazard state-dependent fragility functions (with non-zero initial damage states) are incorporated to calculate the cumulative damages expected after the triggered event with $IM^B$, while accounting for the preceding induced by $IM^A$ (i.e., blue part in Figure 1b, developed in Sect. 2.3). For risk assessment, (4) the incremental loss

obtained from the economic consequence model attached to the classification scheme $T_j^B$ (i.e., replacement costs and related loss ratios per $D_{jy}^B$) is integrated (i.e., yellow part in Figure 1b, explained in Sect. 2.4). These modules are described hereafter.

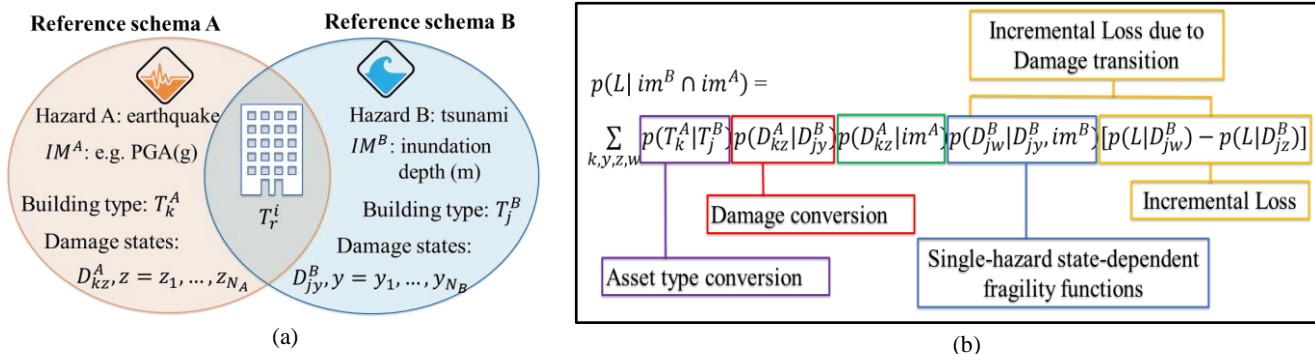

**Figure 1.** (a) Example of the principle proposed for classifying the same building class into two hazard-dependent reference schemes with associated fragility models. (b) Schematic representation of the proposed method that is developed afterward.

## 2.1 Exposure modelling: taxonomic description, inter-scheme conversion and spatial aggregation of building classes

The classified building stock under the first hazard-dependent classification scheme $T_k^A$ is spatially aggregated into a set of geocells that capture the local spatial variations of the hazards' IM of interest across the study area. For such a purpose, we recommend using variable resolution exposure models in the form of Central Voronoi Tessellations (CVT). Besides spatially representing the building portfolio, they also provide a representative IM per geocell for reliable and computationally

efficient vulnerability estimations (Pittore et al., 2020; Gómez Zapata et al., 2021e). They also implicitly serve as common minimum reference units (MRU) aggregation entities between exposure and hazard (Zuccaro et al., 2018). This is because for their derivation, one can consider the combination of local variations of the hazard intensity measures (IM) and certain





exposure proxies (e.g., population density) across the same area. CVT-based models may be useful in a multi-hazard risk
context where the spatial correlation of various IM can differ (e.g., ground-shaking and tsunami inundation) (Gill and
Malamud, 2014).

As shown in Pittore et al., (2018), every building class $k$ that belongs to one scheme $A$ can be described in terms of
basic observable features $\{F\}_m$ within a faceted taxonomy, that is, a building classification schema in which building classes
result from the characterisation of individual attributes, or facets (Brzev et al., 2013; Silva et al., 2018, 2022). This
disaggregation is the common underlying vocabulary to obtain the probability that a building class within the source scheme
$(T_k^A)$ corresponds to another class within the target scheme $(T_j^B)$. As proposed in Gómez Zapata et al., (2022b), the degree of
compatibility between the buildings classes belonging to both schemes can represented by a compatibility matrix $p(T_k^A|T_j^B)$
to account for the uncertainties when there is not a trivial (one-to-one) mapping. Knowing in advance certain exposure metrics
of the source scheme $\{R\}_{T_k^A}$ (i.e., building counts), the respective values of the target scheme $\{R\}_{T_j^B}$ are obtained by applying
the dot product (Eq. 2).

$$\{R\}_{T_j^B} = p(T_k^A|T_j^B).\{R\}_{T_k^A} \qquad\qquad \text{Eq. 2}$$

*2.2 The probabilistic description and compatibility of inter-scheme damage states*

We consider how the fragility functions associated with $T_k^A$ and $T_j^B$ may have diverse numbers and descriptions of
damage states per considered hazard-dependent vulnerability scheme $(D_{kz}^A, z = z_1, \ldots, z_{N_A}$ and $D_{jy}^B, y = y_1, \ldots, y_{N_B})$. To
harmonise their equivalence, we propose obtaining their probabilistic inter-scheme compatibility as a set of matrices
$p(D_{kz}^A|D_{jy}^B)$. This is achieved after having evaluated how the likely observable characteristics linked to each damage state
within $D_{kz}^A$ and $D_{jy}^B$ can be expressed in terms of another one. For this aim, we first propose the use of the AeDES form
(Agibilità E Danno in Emergenza Sismica (usability and damage in seismic emergency)) of the Italian Civil Protection (Baggio
et al., 2007) as a standard scoring system to create a synthetic dataset based on the likely observable damage on individual
building components. Although it was originally proposed for post-earthquake damage data collection, we propose to
transversally use it to describe every damage state $z$ and $y$ of $D_{kz}^A$ and $D_{jy}^B$, respectively. Expert elicitation is used on the
AeDES form to create heuristics evaluating the expected damage extension per building type and damage-limit-states. For this
aim, we make use of its implicit scale within a range of 0=L to 9=A over the building components $n$, (low-level taxonomic
attributes) as shown in Figure 2. We decided to only include four out of these six components that can be found in any building
type as listed in Eq. 3 as stairs and pre-existing damage are not always present in all buildings. The importance of such building
components for assessing their physical vulnerability has been documented in previous studies to ground-shaking (e.g.
Lagomarsino et al., 2021) and tsunamis (e.g. Del Zoppo et al., (2021)).

$$n = \{vertical\ structure\ (VS); floor\ (FL); roof\ (RF); infills\ and\ partitions\ (IP)\} \qquad \text{Eq. 3}$$





| Damage level - extension | DAMAGE [(1)] | | | | | | | | | |
| | D4-D5 Very Heavy | | | D2-D3 Medium-Severe | | | D1 Light | | | Null |
| Structural component Pre-existing damage | $> 2/3$ | $1/3 - 2/3$ | $< 1/3$ | $> 2/3$ | $1/3 - 2/3$ | $< 1/3$ | $> 2/3$ | $1/3 - 2/3$ | $< 1/3$ | |
| | A | B | C | D | E | F | G | H | I | L |
| 1 Vertical structures | ☐ | ☐ | ☐ | ☐ | ☐ | ☐ | ☐ | ☐ | ☐ | ◯ |
| 2 Floors | ☐ | ☐ | ☐ | ☐ | ☐ | ☐ | ☐ | ☐ | ☐ | ◯ |
| 3 Stairs | ☐ | ☐ | ☐ | ☐ | ☐ | ☐ | ☐ | ☐ | ☐ | ◯ |
| 4 Roof | ☐ | ☐ | ☐ | ☐ | ☐ | ☐ | ☐ | ☐ | ☐ | ◯ |
| 5 Infills and partitions | ☐ | ☐ | ☐ | ☐ | ☐ | ☐ | ☐ | ☐ | ☐ | ◯ |
| 6 Pre-existing damage | ☐ | ☐ | ☐ | ☐ | ☐ | ☐ | ☐ | ☐ | ☐ | ◯ |

**Figure 2.** Scale to assess the damage level on buildings as proposed by the AeDES form. Reprinted from Baggio et al, 2007.

A heuristic is generated by scoring the four components in Eq. 3 per damage state, per fragility function, per building class of both exposure classification schemes. This is done through expert elicitation and establishes a training dataset of the possible observable damage extent $\{OD\}_n$ in a harmonized manner. For instance, one set of $\{OD\}_n$ (for a given damage state and building type) is made up by a set of four numbers from 0 to 9, e.g., n = {1, 2, 1, 3}, meaning level I for VS and RF, level H for FL and level G for IP (Eq. 3). Thereafter, using the total probability theorem, the probability that the damage state z of a building class j in a scheme A corresponds to damage state y of building class j in scheme B can be calculated by Eq. 4.

$$p\left(D_{kz}^A | D_{jy}^B\right) = \sum_n p\left(D_{kz}^A | \{OD\}_n \cap D_{jy}^B\right) p\left(\{OD\}_n | D_{jy}^B\right) \qquad \text{Eq. 4}$$

We assume that the representations of damage states within the two considered schemes are conditionally independent (⫫). Thereby, given the information of the scored observable damage on the individual components $\{OD\}_n$, we can describe the source damage scheme $D_{kz}^A$ to be modelled in terms of $\{OD\}_n$ that jointly compose the target scheme $D_{jy}^B$: $D_{jz}^A \perp\!\!\!\perp D_{jy}^B | \{OD\}_n$. Thus, Eq. 4 can be expressed as a product, given by Eq. 5.

$$p\left(D_{kz}^A | D_{jy}^B\right) = \sum_n p(D_{kz}^A | \{OD\}_n) p\left(\{OD\}_n | D_{jy}^B\right) \quad \text{since } D_{kz}^A \perp\!\!\!\perp D_{jy}^B | \{OD\}_n \qquad \text{Eq. 5}$$

We obtain a probabilistic compatibility degree between damage states ($D_{kz}^A, z = z_1, \dots, z_{N_A}$ and $D_{jy}^B, y = y_1, \dots, y_{N_B}$) for every pair of combination of building classes $T_k^A$, and $T_j^B$ through a Bayesian formulation as presented in Eq. 6.

$$p\left(D_{kz}^A | D_{jy}^B\right) = \sum_n p(D_{kz}^A | \{OD\}_n) p\left(D_{jy}^B | \{OD\}_n\right) \frac{p(\{OD\}_n)}{p\left(D_{jy}^B\right)} \qquad \text{Eq. 6}$$

The terms $p(D_{kz}^A | \{OD\}_n)$ and $p\left(D_{jy}^B | \{OD\}_n\right)$ in Eq. 6 can be solved through supervised machine learning techniques for classification (e.g., logistic regression, naive Bayes, decision trees) to predict the probabilities between the training sets





and a synthetic testing dataset. The selection of the machine learning technique, naturally, carries epistemic uncertainties (Mangalathu et al., 2020) whose investigation is beyond the scope of this study. The testing dataset is obtained after generating random numbers of all the possible combinations of the AeDES-based scores. With this dataset we express the conditional

probabilities of having damage states $D_{kz}^A, z$ and $D_{jy}^B, y$ (for each building class within schemes A and B given $\{OD\}_n$. The term $p(\{OD\}_n)$ is a marginal probability that can be assumed to represent the proportion of one observation out of exhaustive combinations of $\{OD\}_n$. Lastly, $p(D_{jy}^B)$ describes the proportion of each damage state $y$ within each building class $k$ in the training dataset for scheme B. Once Eq. 6 is solved, the expression $p(D_{kz}^A|D_{jy}^B)$ is obtained, which stems from the probabilistic inter-scheme damage compatibility matrix for each possible pair of buildings within schemes A and B. After having established

the compatibility between building classes and damage states, a special set of fragility functions is needed to follow the damage progression inflicted by the second hazard. They are explained hereafter.

### *2.3 State-dependent fragility functions*

The next steps of the method are carried out within the reference vulnerability scheme of the second hazard. Let us suppose that the damage state $w$ belongs to $D_{jy}^B, y = y_1, \dots w \dots, y_{N_B}$. Eq. 7 represents the conditional probability that the

building $j$ (of the scheme B) can move to a progressive state $w$ given the action of $IM^B$ when it already presented a damage state $y$ due to the action of $IM^A$. For such a process, it was already classified in terms of scheme B, thanks to the compatibilities between damage states from different scales described above.

$$p(D_{jw}^B|D_{jy}^B, IM^B) \qquad \text{Eq. 7}$$

The former expression defines a probabilistic state-dependent fragility function composed of transition probabilities between increasing damage states. For instance, for the scheme B, this description follows: $y_{N_B} - y_{N_B-(N_B-1)}; y_{N_B} -$

$y_{N_B-(N_B-2)} \dots$ For a fragility model $D_{T_r^i}$ designed for a set of building types $T_r$, and composed of $q_{N_i}$ damage states (for any hazard of interest i), the required set of transition probabilities for a given range of hazard intensities are completely defined by a triangular number $G_f$ as expressed in Eq. 8.

$$G_f = \sum_{D_{T_r^i}=1}^{q_{N_i}} D_{T_r^i} = \frac{(1+q_{N_i})q_{N_i}}{2} \qquad \text{Eq. 8}$$

If a fragility function is composed of $q_{N_i} = 3$ damage states (excluding damage state 0, equivalent to no damage), the required triangular number to express the damage state transitions is $G_f = 6$ (3 from 0; 2 from 1: 1 from 2). A visual example

of such transition probabilities within fragility functions for several hazard-dependent models (also including $T_k^A; T_j^B$ and their respective sets of damage states $z_{N_A}; y_{N_B}$ is presented in Figure 3. For each $T_r^i$, it is then necessary to determine the




probabilistic representation of the damage state transitions in every $G_f$ that forms the damage-state fragility functions. For that aim, the relation proposed in Mignan et al., (2014) expressed in Eq. 9 is adopted.

$$\delta_{w|y} = \frac{1}{2} erfc \left( \frac{-ln(d_w) - \mu_{w|y}}{\sigma_w \sqrt{2}} \right)$$

Eq. 9

This relation proposes that the lognormal cumulative distribution $\delta_{w|y}$ be used to find a damage state $w$ conditional on the previous occurrence of damage state $y$. It describes the conditional transition probabilities of exceedance between damage states. Every mean value of the damage transitions $\mu_{w|y}$ of $D^B_{jy}$ can be expressed as $\mu_{w|y} = \mu_w - \varphi ln(d_y)$, where $d_w$ and $d_y$ are the corresponding IM to the damage states $w$ and $y$, and $\varphi$ represents the scaling factors that modify the fragility function and shifts to higher probabilities of exceeding a given damage state for the same hazard intensity $IM^B$. The scaling factors $\varphi$ are obtained as ad-hoc calibration parameters, similarly as proposed in Mignan et al., (2014). The fragility functions used to constrain the state-dependent fragility functions should have been derived only for the actual second acting hazard (i.e., far-field tsunamis). Thus, the use of those derived analytically is advised over empirical ones (which had implicit the damaged induced by ground-shaking in their derivation).

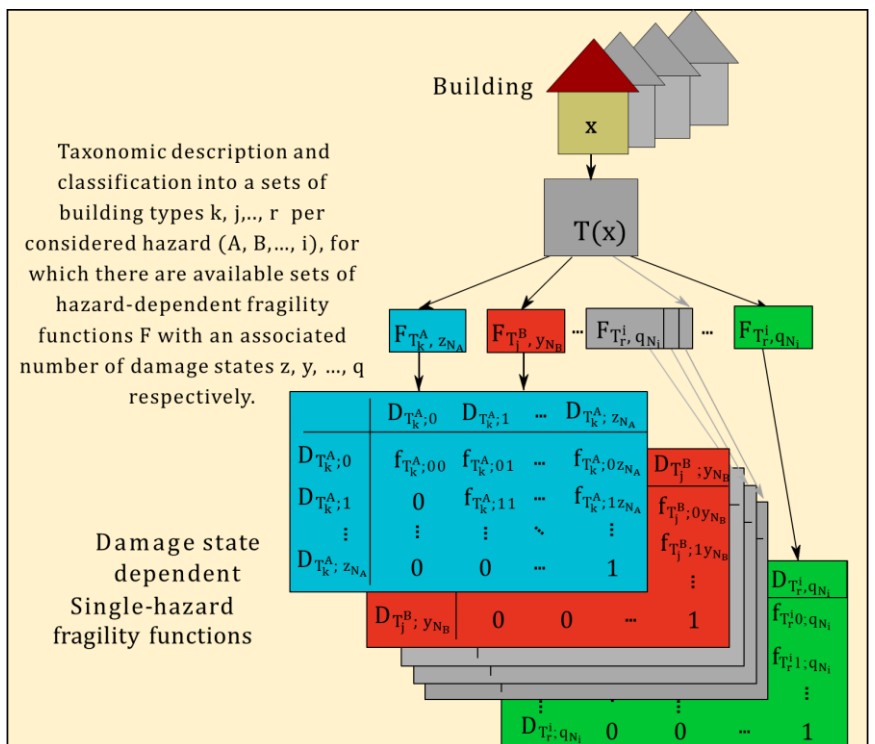

**Figure 3.** Example of a set of damage state-dependent fragility functions for several single hazard fragility functions comprising progressive transition probabilities. Figure modified from Gómez Zapata et al., (2020).





### 2.4 *Loss assessment for sequences of cascading hazards scenarios*

We propose a simple economical consequence model that assigns the replacement cost ratios to every damage state of the building classes $T_j^B$. The incremental economic loss, defined as the difference in the expected loss resultant from the initial damage state and final damage state, is calculated in terms of the reference scheme $B$ as:

$$p\left(L|D_{jw}^B\right) - p\left(L|D_{jz}^B\right) \qquad \text{Eq. 10}$$

Combining the two inter-scheme compatibility matrices, ($p\left(T_k^A|T_j^B\right)$ and $\left(D_{kz}^A|D_{jy}^B\right)$, along with Eq. 7 and Eq. 10, we obtain the formulation in Eq. 11, which is identical to the one in Figure 1b. This allows us to calculate the probability of observing an incremental loss due to the cumulative damage during the sequence of hazard-scenarios.

$$p(L|IM^B \cap IM^A) = \sum_{k,y,z,w} p\left(T_k^A|T_j^B\right)p\left(D_{kz}^A|D_{jy}^B\right)p\left(D_{kz}^A|IM^A\right)p\left(D_{jw}^B|D_{jy}^B, IM^B\right)\left[p\left(L|D_{jw}^B\right) - p\left(L|D_{jz}^B\right)\right] \qquad \text{Eq. 11}$$

Eq. 11 represents the disaggregated loss caused by the triggered event upon the buildings with a pre-existing damage (induced by $IM^A$). Finally, the likely loss for the entire sequence can be obtained by summing up Eq. 1 and Eq. 11.

## 3 Application example

### 3.1 *Context of the study area: Metropolitan Lima, Peru*

In 2022, Peru had a population of around 33 million people, with nearly 58% of this living in coastal communities. In Løvholt et al., (2014) it was stated that this country has the largest population exposed to tsunamis in the American continent. Lima, its capital, with nearly 10 million inhabitants (around one third of the country's population) is home to the most important
political, industrial, and economic activities of the country. Lima ranks as the capital city exposed to the highest seismic hazard in South America (Petersen et al., 2018), and as the second city in the world in terms of the value of working days lost relative to the national economy due to earthquakes (Schelske et al., 2014). This city has suffered devastating disasters in the past. For instance, in 1586 and 1724 earthquakes triggered tsunami run-ups over 24 m (Kulikov et al., 2005). The 1746 earthquake, with an estimated magnitude of $M_w$ 8.8 (Jimenez et al., 2013), produced a tsunami with local height of 15 to 20 m (Dorbath et al.,
1990) and destroyed the city. In 1974, a $M_w$. 8.1 event produced widespread damage and caused losses of ~ 7.5 billion dollars. Since then, the city has been experiencing continuous urbanization with generally poor structural design (Tarque et al., 2019).

The 1746 earthquake for scenario for earthquake and tsunami modelling was also used in Adriano et al., (2014) to estimate the damage probabilities of the residential building stock of Callao (part of the Metropolitan area of Lima) using the empirical tsunami fragility functions of (Suppasri et al., 2013) for four building types. More recently, Ordaz et al., (2019)
developed probabilistic earthquake and tsunami risk forecasts for Callao. However, that study did not describe the vulnerability models used, nor the method employed to address the non-linear damage accumulation. To the authors' best knowledge, neither





cumulative damages due to earthquake and tsunami scenarios nor the use of analytical tsunami fragility functions for Lima have been reported in the scientific literature.

### 3.2    *Scenarios of earthquake and tsunami for Lima*

260        We use the dataset compiled by Gómez Zapata et al., (2021e) which is composed of six earthquakes with moment magnitudes ranging from 8.5 to 9.0 $M_w$, which were made available in Gómez Zapata et al., 2021c). In that dataset, each event is represented by an associated 1,000 realisations of cross-correlated ground motion fields (GMF) for peak ground acceleration (PGA) and spectral accelerations at 0.3 and 1.0 seconds. The selection of these spectral periods depends upon the fragility function's IM associated with the building classes of the exposure model (Sect. 3.3). The simulated GMF were obtained making

use of the ground motion prediction equation (GMPE) proposed in Montalva et al., (2017) and the spatially cross-correlation model of Markhvida et al., (2018) employing the OpenQuake Engine (Pagani et al., 2014). For the site term of the GMPE, the dataset reported in Ceferino et al., (2018), which combined the slope-based $Vs_{30}$ values of Allen and Wald, (2007) and a seismic microzonation (Aguilar et al., 2013) was used. On the tsunami modelling side, we reuse the data repository Harig and Rakowsky, (2021) that compiles tsunami inundations for each of the mentioned six earthquakes using the finite element model

TsunAWI. Similarly as performed by Harig et al., (2020), the inundation values were interpolated to a raster file with grid cell dimensions of 10×10 m. Figure 4 shows three of the tsunami inundation scenarios for the study area.

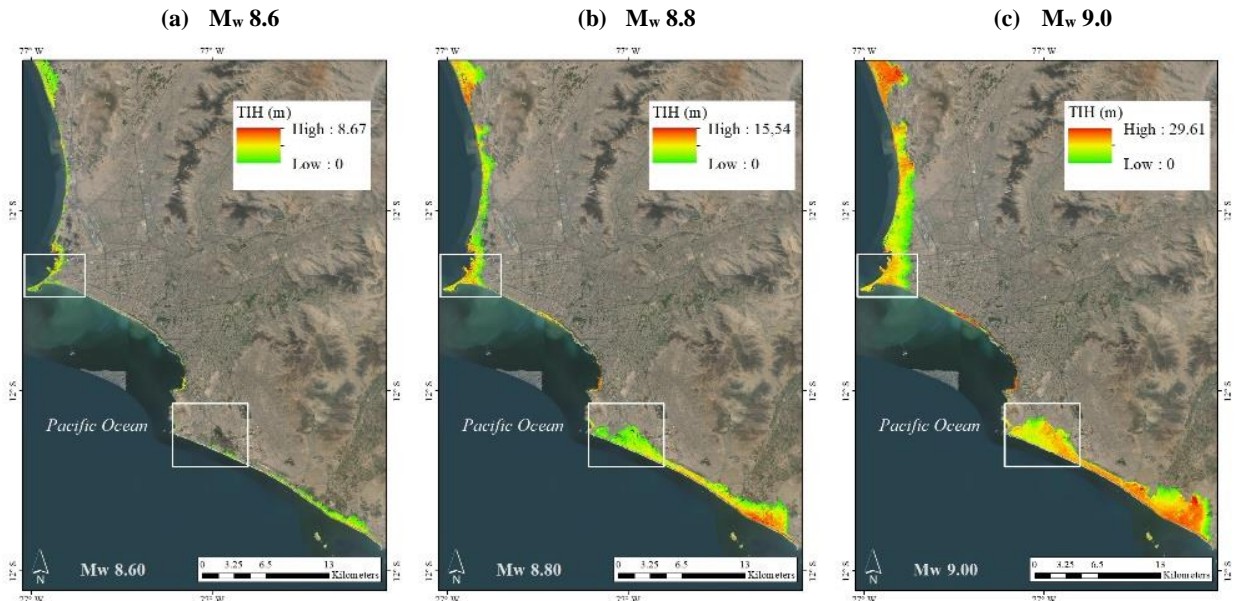

**Figure 4.** Expected tsunami inundation heights for three out of the six considered scenarios per moment magnitude ($M_w$). These raster products are available from Harig and Rakowsky, (2021). Two densely populated areas are depicted by white rectangles: in the north the "La Punta" (Callao) and Chorrillos in the south. Updated figure from Gómez Zapata et al., (2021e). Map data: ©Google Earth 2021.





## 3.3    *Exposure modelling: taxonomic description, inter-scheme conversion and spatial aggregation of building classes for Lima*


We make use of the existing building exposure models that represent the residential building stock of Metropolitan Lima for ground shaking vulnerability that were developed by Gómez Zapata et al., (2021e) and are available from Gómez Zapata et al., (2021b). Such a building classification was defined by relating some covariates included within the last official

Peruvian census from 2017 (INEI, 2017) at the block-level with respect to 21 classes proposed by the South American Risk Assessment (SARA) project (Yepes-Estrada et al., 2017) through a mapping scheme proposed from expert-elicitation (GEM, 2014). Since that information was provided for dwellings, the so-called "dwelling ratios" proposed by SARA were also implemented to obtain the building counts per class. A description of these building classes is presented in Table 1.

**Table 1. SARA Building classes proposed for Metropolitan Lima and Callao with their replacement costs as reported in Yepes-Estrada et al., (2017). The intensity measures (IM) of their associated fragility functions, as reported in Villar-Vega et al., (2017), are also provided.**

| SARA building classes in Lima and Callao | Description | IM | Repl. Cost (USD/bdg) | Building counts |
|---|---|---|---|---|
| MUR-H1-3 | Unreinforced masonry, between 1–3 stories | PGA | 18,000 | 248799 |
| MUR-ADO-H1-2 | Unreinforced masonry (MUR) with adobe (ADO), 1–2 stories | PGA | 15,000 | 209837 |
| MUR-STDRE-H1-2 | Dressed stone (STDRE) unreinforced masonry, 1–2 stories | PGA | 15,000 | 209837 |
| W-WBB-H1 | Wood (W), bamboo (WBB), 1 story | S.A at 0.3s | 12,000 | 187355 |
| W-WWD-H1-2 | Wood, bahareque and Quincha (WWD, wattle and daub construction), 1–2 stories | S.A at 0.3s | 15,000 | 149884 |
| W-WS-H1-2 | Wood, solid wood (WS), 1–2 stories | S.A at 0.3s | 12,000 | 127401 |
| W-WLI-H1-3 | Wood (W), light wood (WLI), 1–3 stories | S.A at 0.3s | 31,500 | 123654 |
| ER-ETR-H1-2 | Reinforced (ETR) rammed earth (ER), 1–2 stories | PGA | 15,000 | 89931 |
| MUR-STRUB-H1-2 | Unreinforced masonry with rubble (field stone) or semi-dressed stone, 1–2 stories | PGA | 15,000 | 89931 |
| W-WHE-H1-3 | Wood, Heavy wood (WHE), 1–3 stories | S.A at 0.3s | 12,000 | 82436 |
| MCF-DNO-H1-3 | Confined masonry (MCF), non-ductile, 1–3 stories | PGA | 40,500 | 66749 |
| MCF-DUC-H1-3 | Confined masonry, ductile, 1–3 stories | PGA | 126,000 | 66749 |
| MR-DUC-H1-3 | Reinforced masonry, ductile, 1–3 stories | PGA | 360,000 | 16745 |
| CR-LFINF-DNO-H1-3 | RC with infilled frame, low rise (non-ductile) | PGA | 126,000 | 13925 |
| UNK | Unknown | S.A at 0.3s | 12,000 | 8432 |
| CR-LFINF-DUC-H1-3 | RC with infilled frame, low rise (ductile) | PGA | 288,000 | 7519 |
| CR-LDUAL-DUC-H4-7 | Reinforced concrete (RC) with dual wall system, medium rise (ductile) | S.A at 1.0s | 1,080,000 | 125 |
| CR-LWAL-DNO-H4-7 | RC wall system, non-ductile, 4–7 stories | S.A at 1.0s | 472,500 | 76 |
| CR-LWAL-DUC-H4-7 | RC wall system, ductile, 4–7 stories | S.A at 1.0s | 1,080,000 | 76 |
| CR-LWAL-DUC-H8-19 | RC wall system, ductile, 8–19 stories | S.A at 1.0s | 3,456,000 | 34 |
| CR-LDUAL-DUC-H8-19 | RC with dual wall system, high rise (ductile) | S.A at 1.0s | 3,456,000 | 32 |

It is worth noting that although these typologies are similar to those of the SARA exposure model, there are differences between the building counts reported by that project and our model. This might be due to the vintage of the input census datasets (2007 vs. 2017, respectively), the thematic detail induced by the spatial aggregation entities (districts/ blocks/ CVT),
having merged some building classes in terms of similar heights, and having reduced the number of unknown (UNK) type (~91% with respect the SARA model). The resultant exposure model is made up of ~1,657,635 residential buildings, a 25% increase with respect the SAA model. However, as observed in Gómez Zapata et al., (2022b), this scheme does not properly capture the presence of high-rise buildings, underestimating their presence while overestimating the wooden types.

These SARA buildings are spatially aggregated onto Central Voronoi Tessellations (CVT) to form seismic-oriented
exposure models. It is worth noting that the construction of such heterogeneous aggregation units was based on the selection of an underlying distribution that spatially combined and normalised two weighted map layers, namely: (1) a tsunami inundation depth from a Mw 9.0 scenario (70% weight), and (2) the population density at the block level (30%). The resulting model provides higher resolution cells where both conditions are maximised whilst coarser geocells occur when one can expect their absence. Further details about these models are available in Gómez Zapata et al., (2021a, b). Figure 5 shows the percentage
of building typologies grouped by their main structural materials expected within each geocell.

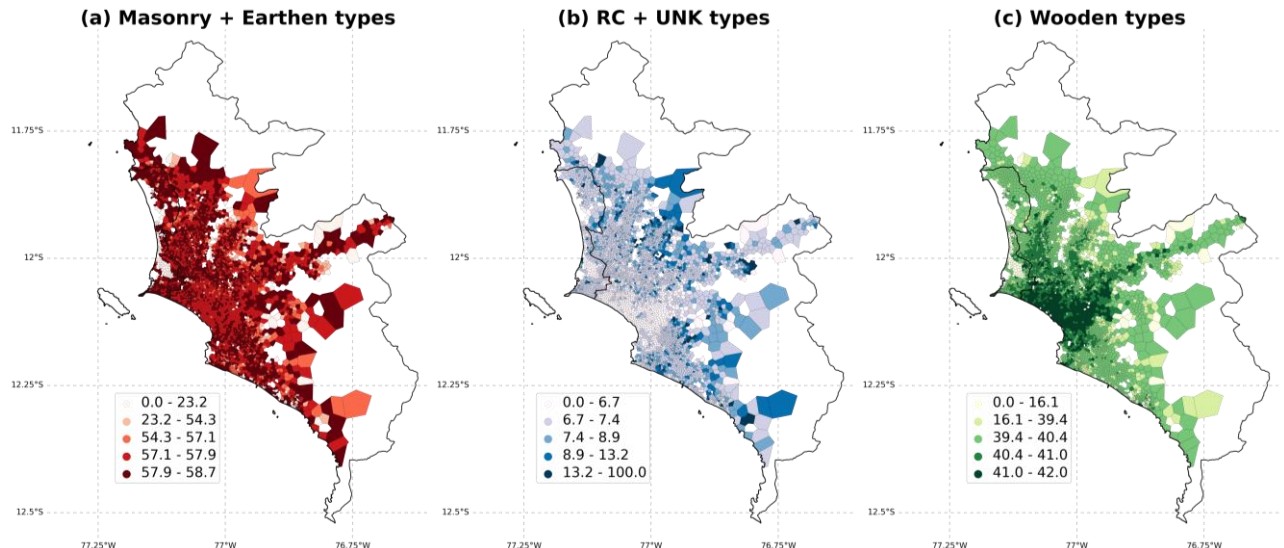

**Figure 5.** Spatial distribution of the percentage of the main structural material of the residential buildings in Metropolitan Lima in each CVT geocell using the dataset of Gómez Zapata et al., (2021b). The colour scale represents the material type (red: masonry and earthen; blue: reinforced concrete and UNK; green: wooden types). Only CVT that intersected the census-based blocks (INEI, 2017) are shown.

The analytically derived set of seismic fragility functions by Villar-Vega et al., (2017) are assigned to every SARA class. They will be used to obtain the damage distribution for the cross-correlated ground motions per earthquake scenario (Sect. 3.2). For this vulnerability assessment, we use the replacement cost as given by Yepes-Estrada et al., (2017a), presented in Table 1. For their damage states, we assumed loss ratios of 2%, 10%, 50%, and 100%, respectively.





On the tsunami vulnerability side, we represent the commonly exposed residential building stock to earthquakes and
tsunamis in terms of two classification schemes, namely the Suppasri et al., (2013) and Medina, (2019) schemes which provide
sets of empirical and analytical fragility curves, respectively. The former one was made available for Lima in Gómez Zapata
et al., (2021b) and is comprised of six typologies. Notably, its corresponding set of empirical tsunami fragility functions (with
six damage states) was derived by implicitly addressing the damage induced by the ground-shaking after the $M_w$ 9.1 2011
Japan earthquake and tsunami. Due to this reason, the steps outlined in Sections 2.2 and 2.3 are not developed for the Suppasri
et al. (2013) scheme. Their related direct scenario-based loss estimates were reported in Gómez Zapata et al., (2021e) from the
variations obtained from seven geographical entities used to spatially aggregate the residential building portfolio of Lima, and
presented in Sect. 4 for comparative purposes in contrast with the offered method applied to the Medina (2019) scheme. This
second type of classification is to the authors' knowledge the only available model that provides analytical far-field tsunami
fragility functions for the South American Pacific Coast. It includes six typical buildings located in Tumaco (Colombia)
initially defined in Medina, (2019), which are generalized in this study. They are M-PN (wooden), M-MP (masonry), M-PCP1-
T1 (framed RC, one storey with similar length-width ratio), M-PCP1-T2 (framed RC, one storey, with a higher length to width
ratio), M-PCP2 (framed RC, 2 storeys), and M-PCP3 (framed RC, 3 or more storeys). Their associated set of fragility functions
was developed following the method proposed in Medina et al., (2019) to define the structural fragility due to tsunami forces.
A summary that regards the structural characteristics of these building types and the method adopted in deriving these models
are provided in the data repository Gómez Zapata et al., (2022a).

As explained in Sect. 2.1, every building class within the three schemes of interest is disaggregated into attributes
within the GEM v.2.0 faceted taxonomy. As done in Gómez Zapata et al., (2022b), fuzzy compatibility levels between the
attribute values and building classes are assigned through expert elicitation. Thereby, synthetic surveys based on the possible
combinations of attributes that every building class may describe are employed to solve the compatibility scores and obtain
the probabilistic inter-scheme compatibility matrices in the form of $p\left(T_k^A \middle| T_j^B\right)$. Subsequently, we can obtain the building
counts under the tsunami classification scheme. This is done considering the SARA classification (Figure 6a), as the source
scheme $\{R\}_{T_k^A}$ and the inter-scheme conversion matrix (Figure 6b). Then, the corresponding counts under the tsunami scheme
of Medina (2019) $\{R\}_{T_j^B}$ (Figure 6c) are obtained by applying a dot product (Eq. 2).

The inter-scheme conversion between SARA and the Suppasri et al., (2013) classes for Lima was reported in Gómez
Zapata et al., (2021e). The replacement costs values of the building classes within the Medina (2019) scheme are assumed to
be the same as the SARA class for which the largest compatibility value was obtained from the inter-scheme compatibility
matrix (Figure 6-b). We have adopted identical loss ratios per limit damage state as the ones assumed for earthquake
vulnerability. Similar loss ratios were also adopted in Antoncecchi et al., (2020) to assess the vulnerability of buildings to
tsunamis using empirical fragility functions. It is worth noting that only the commonly exposed buildings to each pair of hazard
scenarios (i.e., intersection between the IM of Figure 4 and Figure 5) are considered for the assessment of cumulative damage
after the cascading sequence.




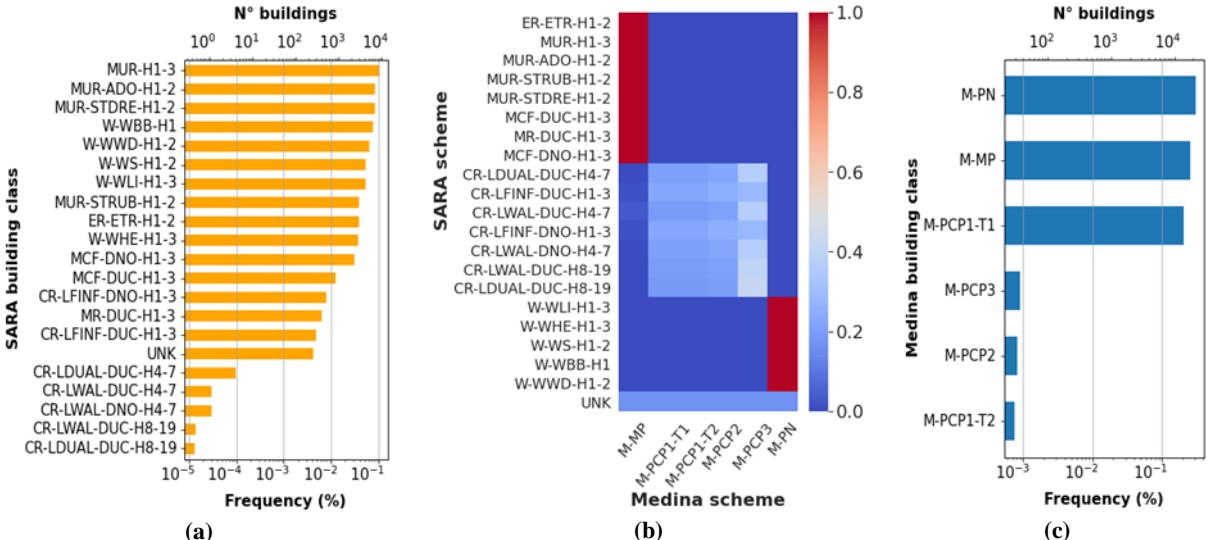

**Figure 6.** Classification of the buildings in the maximum exposed area to both perils (M$_w$ 9.0 scenario) in terms of the (a) seismic-vulnerability oriented SARA classes (used as a source scheme) and (b) the inter-scheme conversion matrix. The former two models are used as inputs to obtain the (c) proportions for the tsunami-oriented building classes of Medina (2019).

### 3.4    The probabilistic description and compatibility of inter-scheme damage states for Lima

We obtain the inter-scheme damage compatibility matrices, $p\left(D_{kz}^A \middle| D_{jy}^B\right)$, following the method presented in Sect. 2.2 to probabilistically harmonise the damage states that define the fragility functions of $A$ (SARA) and $B$ (Medina). It is worth noting that although $A$ and $B$ comprise four damage states they do not have a trivial equivalence. $A$ defines a single damage criterion for the entire set of building classes closely following the proposal by Lagomarsino and Giovinazzi, (2006) as a function of the yielding and ultimate spectral displacements. Conversely, $B$ uses a building class-dependent parametrization based on the HAZUS inter-storey drift ratios to define the structural damage levels on pre-code structures.

First, we use the AeDES scale to score the admissible observable damage extension on individual building components ($n$ in Eq. 3) through expert elicitation, which can jointly describe each building-specific damage states of every scheme's fragility functions ($D_{kz}^A$, $D_{jy}^B$). Some examples of this procedure are displayed in Figure 7. These heuristics configure the training datasets. Subsequently, we have configured the testing datasets. They are composed of a synthetic dataset of 10,000 exhaustive possible combinations of the observable AeDES score-based damage extension $\{OD\}_n$. Thereby, the likelihood terms and $p\left(D_{jy}^B \middle| \{OD\}_n\right)$ in Eq. 6 represent the probability of classifying each damage state $D_{kz}^A$ and $D_{jy}^B$ given the set of scored building components $\{OD\}_n$.


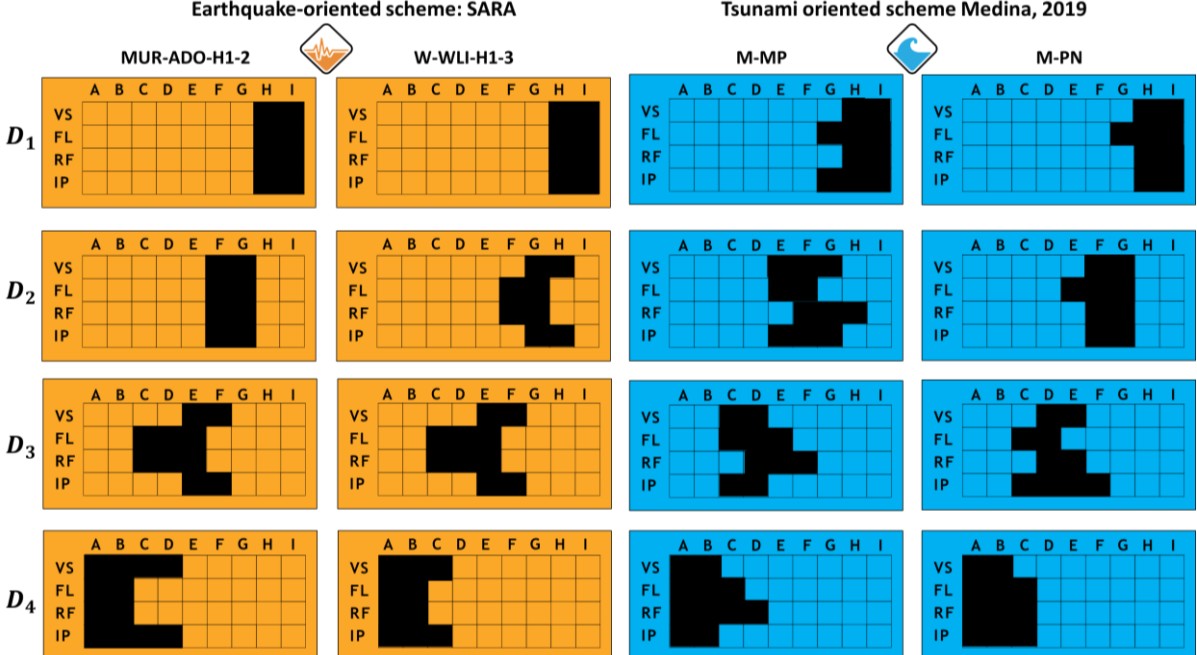

**Figure 7.** Examples of the AeDES-based heuristics (see Figure 2 and Eq. 3) used to describe the expected observable damage features per damage state per building class within two hazard-dependent vulnerability schemes.

To obtain the likelihood terms of in Eq. 6, we have decided to use the Gaussian Naïve Bayes supervised machine-learning classification-algorithm. It is available in the free software library Scikit-learn for the Python programming language (Buitinck et al., 2013). This selection is suitable for our classification problem because the observable damage heuristics can be assumed as normally distributed continuous data. This can be observed from the heuristic shown in Figure 7 where the central damage states (i.e., moderate and extensive) show wider ranges of combinations of observable damage with respect to the lowest (slight) and largest (collapse) states. For illustrative purposes, in Figure 8 we show one of the possible sets for the likelihood probabilities predicted for each damage state described in terms of observable damage extension with respect to the AeDES scale upon two building components (VS, IP) for two material-based typologies in the commonly exposed area to both perils, i.e., masonry and wooden structures (see Figure 6a,c).

The marginal probability in Eq. 6, $p(\{OD\}_n)$, is assumed to be the proportion between one observation and the exhaustive combinations (1/10,000). Thereafter, we have obtained the probabilistic inter-scheme damage matrix $p\left(D_{kz}^A|D_{jy}^B\right)$ for each combination of building types from the two schemas (i.e., 21 SARA classes by 6 Medina classes = 126 conversion matrices). Examples of the inter-scheme damage matrices are shown in Figure 9 for three pairs of building types that had the highest inter-scheme compatibility values in Figure 6b. Each of the 126 matrices that relates the damage states for each possible combination of building classes from the two schemas is subsequently weighted by the corresponding value of $p\left(T_k^A|T_j^B\right)$, that is, by the probability of the building classes of the two schemas actually being descriptive of the same actual building (i.e.,


Figure 6-b). When considered in Eq. 11, the damage related matrices are maximized by the most compatible pairs of inter-schema building matrices. The scripts, heuristics, the final set of likelihood distributions, and the related set of compatibility

380    matrices are provided in Gómez Zapata et al., (2022c).

**Figure 8.** Predicted likelihood probabilities of classifying each damage state $D_{jz}^A$ (SARA classes) and $D_{ky}^B$ (Medina classes) given the combinations of observable damage $\{OD\}_n$ in terms of the AeDES scale for mansonry (a, c) and wooden (b, d) buildings.
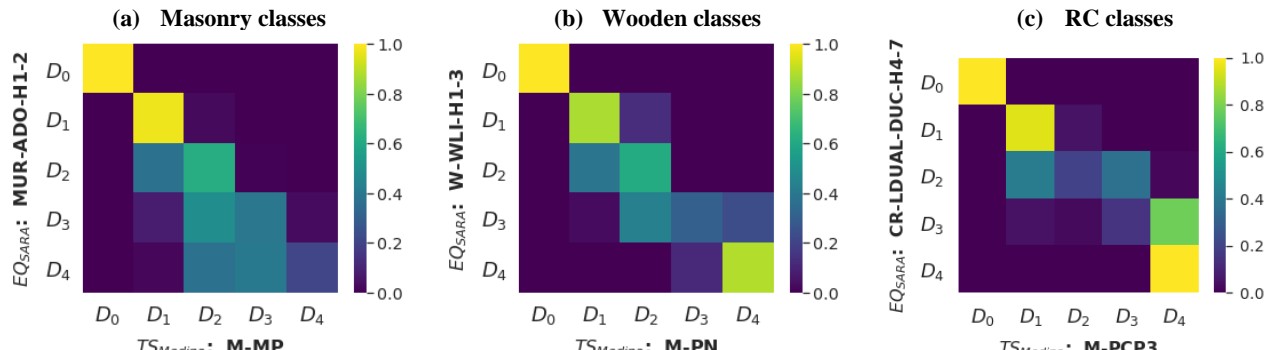

**Figure 9.** Probabilistic inter-scheme damage compatibility matrices for three pairs of building classes whose fragility functions are comprised within the source ($D_{kz}^A$ SARA (EQ: earthquake-oriented) and the target $\left(D_{jy}^B\right)$ Medina (TS: Tsunami-oriented) vulnerability schemes.

### 3.5  *Tsunami state-dependent fragility functions for Lima*

We have followed the method presented in section 2.3 to configure the state-dependent fragility functions based on Scheme $B$ (Medina) with associated analytical far-field tsunami fragility functions. The parameters that define the lognormal cumulative distributions for the four original damage states (assuming an initial undamaged state), and well as for the set of $G_f = 10$ transitions probabilities (from Eq. 8) are provided in the data repository Gómez Zapata et al., (2022a). Figure 10 shows the analytical tsunami fragility functions (continuous lines) and state-dependent fragility curves with their respective damage-transitions (non-continuous lines) for the six building classes.

From Figure 10 it is possible to observe some features of the tsunami damage-state fragility functions based on ad-hoc calibration parameters (Sect. 2.3). For example, the masonry buildings class is the one most fragile to tsunami forces when in an undamaged state. Consequently, their associated state-dependent fragilities are shifted towards the left side of the plot in quite an extreme fashion (Figure 10-a). This means that for that building type there is a higher probability for it to follow a longer damage progression after having been strongly affected by the seismic ground-shaking (dotted and dashed lines). Conversely, for the wooden buildings (Figure 10-b) these are more likely to follow a damage progression than other classes if they were slightly affected by the shaking (see dashed lines). For the two one-storey RC building types assessed (M-PCP1-T1 & M-PCP1-T2) there are negligible differences between the transition probabilities $D_2$-$D_3$ and $D_3$-$D_4$, as well as between $D_1$-$D_3$ and $D_2$-$D_4$. Notably, the inter-distances between these pairs of sets (of damage states) are of a similar order as the ones comprised by one and two damage state(s) respectively. This feature is not present for the other RC buildings with increasing heights nor the wooden types. This observation is dependent on the specific analytical fragility models used and the assumptions adopted to derive them (Eq. 10) and no generalization should be done until it can be further validated through other means.


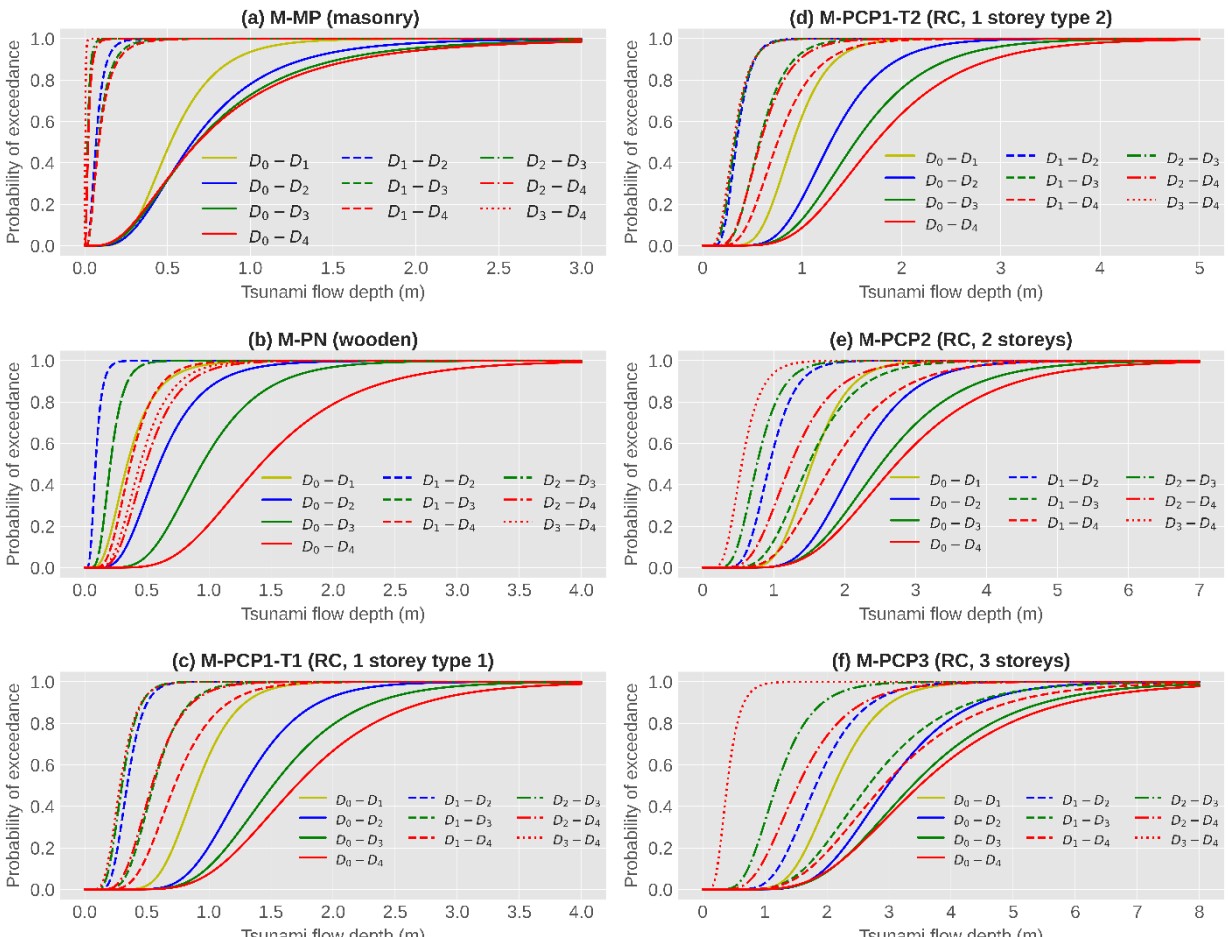

**Figure 10.** Analytical tsunami fragility functions with initial undamaged state as proposed by Medina, (2019) (continuous lines) and derived state-dependent fragility curves (non-continuous lines) for six building classes in terms of flow depth (m) as IM.

## 3.6 *Cumulative damage from consecutive ground shaking and tsunami scenarios in Lima*

The spatially cross-correlated ground motion fields (Sect. 3.2, Fig. 11-a, b), along with the exposure model for seismic vulnerability and their corresponding fragility functions (Sect. 3.3, Fig. 11-d, e) are the first set of inputs required by the engine DEUS (Brinckmann et al., 2021) to estimate the damage distribution and direct economic losses for the residential building stock of Lima after each earthquake scenario. DEUS is a software designed to compute scenario-based risk from any type of natural hazard over spatially aggregated building portfolios. This version of DEUS is an open-source Python program whose number of executions are proportional to the consecutive risk scenarios.

As shown in f, g, the resulting damaged exposure model (after ground-shaking) is used as input for a second execution to account for the cumulative damage induced by the tsunami scenarios. DEUS makes use of the two sets of inter-scheme



compatibility matrices for buildings (Sect. 3.3) and damage states (Sect.3.4) to change from the source earthquake reference scheme to the target tsunami reference scheme (see Fig. 11-g). These are inputs together with the tsunami inundation heights

420 (Sect. 3.2, Fig. 11--c), and state-dependent tsunami fragility functions (Sect. 3.5, Fig. 11-h) for the second run of DEUS. This time, the damage states are updated in the building exposure model, delivering only the disaggregated damage and losses expected from the tsunami. Finally, the cumulative distribution of losses is obtained by adding the latter disaggregated tsunami losses with the initial results derived from the earthquake ground-shaking.

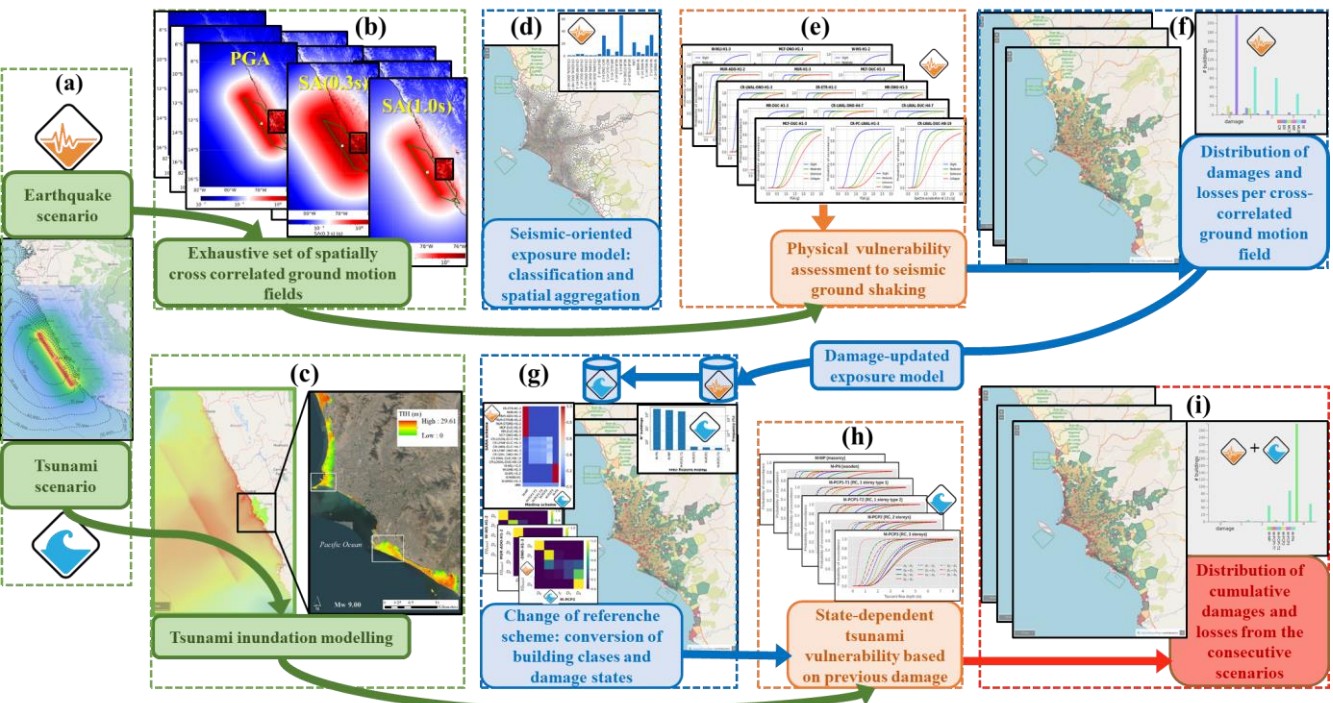

425 **Figure 11.** Proposed workflow for multi-risk assessment in Lima from each pair of consecutive earthquake and tsunami scenarios. A $M_w$ 8.8 event is displayed as an example. The processes regarding the natural hazardous events are highlighted in green. Blue and orange indicate the exposure and vulnerability processes, respectively. The spatially cross-correlated ground motion fields and an initial exposure model (with earthquake-oriented classes) are input for the seismic vulnerability process, which provides the damage-updated exposure models. After the reference scheme conversion processes (building classes and damage states), these sets of damaged-exposure models are 430 inputs for the state-dependent tsunami fragility functions to finally obtain the distribution of cumulative damages and losses (red). Map data of subplot (c) is from ©Google Earth 2021.The basemap and data of suplots (d), (f), (g), (i) are from © OpenStreetMap contributors 2021. Distributed under the Open Data Commons Open Database License (ODbL) v1.0)

## 4    Results

The generated results are presented in the form of loss exceedance curves in Figure 12. This figure reports the

435 probability of exceeding the selected loss metric (replacement cost in USD) for the six earthquake and tsunami scenarios that might impact the portion of the residential building stock of Lima that is commonly exposed to each pair of hazard scenarios. This figure shows five sets of curves, hereby described:




1. Earthquake ground-shaking-induced loss (blue curves). They represent the direct losses due only to seismic ground shaking using the SARA scheme (Villar-Vega et al., 2017). They are obtained through 1,000 realisations of cross-correlated seismic ground motion fields using the models described in section 3.2.

2. Losses obtained from the sole use of empirical fragility functions as simulating a near field tsunami (red curves). These curves represent the losses from the cumulative effects of the shaking and the tsunami (without any possibility to separate both effects). Such losses prediction may be biased since the empirical fragility functions of (Suppasri et al., 2013) assuming an initial undamaged state ($D_{k0}^A$) has not been validated for smaller or larger events. Similarly as it was concluded in Gómez Zapata et al., (2021), we have also observed that as the earthquake magnitude increases, the differences between the two largest loss values in the curve (from the two finest resolution entities) are reduced.

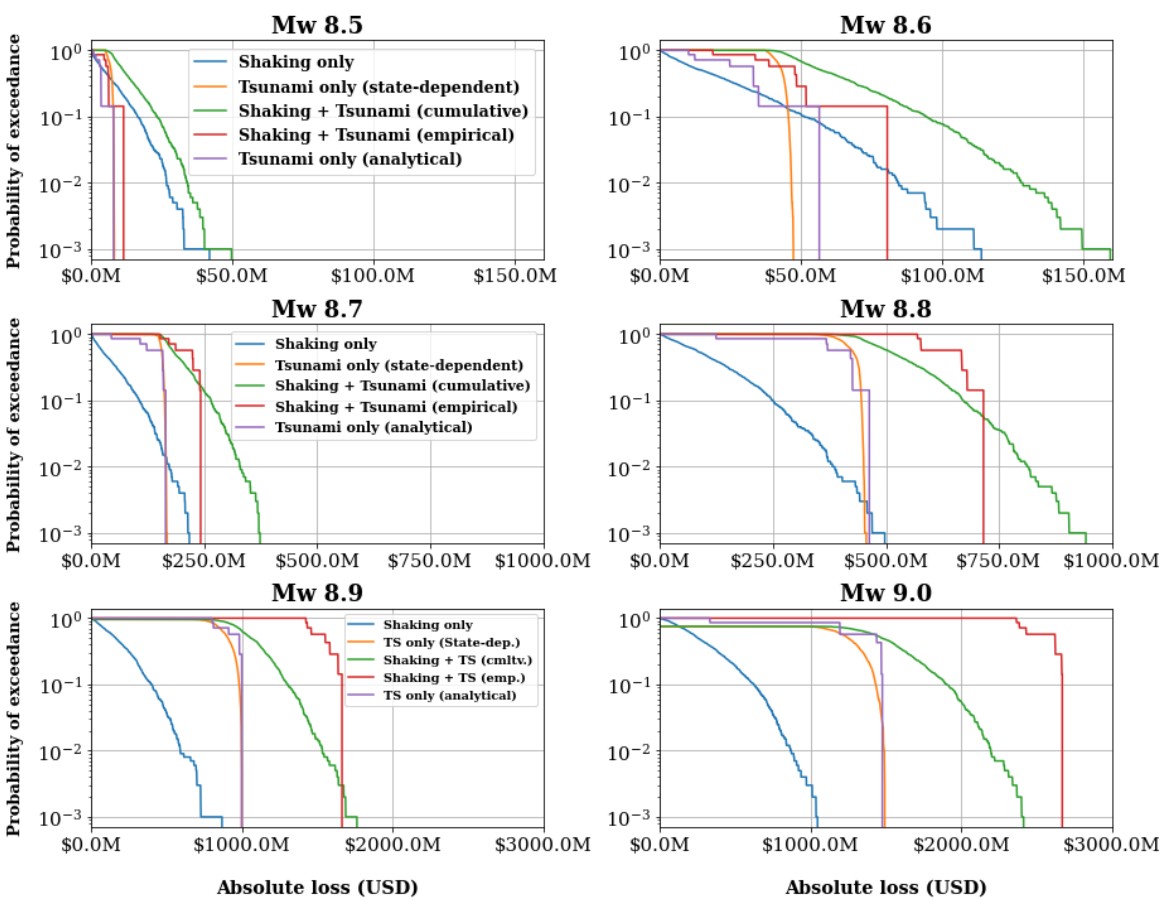

**Figure 12**. Five loss exceedance curves for the residential building portfolio of Lima are presented in each subplot per earthquake magnitude scenario (Mw 8.5-9.0). Three out of the five curves represent the disaggregated losses per hazard event: shaking-induced losses only (blue); far-field tsunami-induced losses (initial undamaged state, purple); state-dependent tsunami-induced losses (with pre-existing shaking induced damage, orange). The green curves represent the losses expected from the cascading sequence whilst the red ones show the losses derived solely using empirical tsunami fragility functions.



3.  Losses obtained from the sole use of analytical fragility functions as simulating a far-field tsunami (purple curves.
    They represent the direct losses obtained solely through the implementation of the analytical tsunami fragility tsunami
    Medina, (2019), while assuming an initial undamaged state ($D_{k0}^A$), thus, neglecting seismic ground-shaking. Similarly,
    as done for the former case (empirical functions), the reduced variability of these results was accounted for through
    computations using seven exposure models, with variable spatial resolutions obtained from a recent study (Gómez
    Zapata et al., 2021e). This is a result of the lack of variability in the seismogenic parameters to vary only the $M_w$, and
    not having assumed distributions for slip-rates, but single values.

4.  The losses related to the tsunami event obtained after using our method (state-dependent fragility functions, orange
    curves). They represent the direct losses which were only derived from the updated exposure model (i.e., with non-
    zero damage states)). This means that these curves only represent the tsunami-induced losses for buildings that have
    already experienced earthquake-related damage. These loss exceedance curves are constructed using Eq. 11. Thus,
    this procedure implied the inter-scheme building conversion $p\left(T_k^A \middle| T_j^B\right)$ derived in section 3.3, the inter-scheme
    damage state conversion $p\left(D_{kz}^A \middle| D_{jy}^B\right)$ obtained in Sect. 3.4, as well as the state-dependent tsunami fragility functions
    constrained in Sect. 3.5.

5.  Cumulative losses (our method) induced by the ground-shaking and tsunami sequence (green curves). They represent
    the losses obtained by adding the shaking-induced losses (blue curves) with the aforementioned disaggregated
    tsunami-induced losses (orange curves), that is, the outcome of the method proposed in this paper. These green curves
    represent, according to our approach, the likely losses that would be expected from each magnitude-dependent
    scenario-based cascading sequence over the considered building stock.

Hereafter we describe some observations that arise from Figure 12.

1.  The resultant losses obtained after having used the two sets (empirical or analytical) of tsunami fragility functions
    (while assuming initial undamaged states) are profoundly different. As expected, the use of the empirical tsunami
    fragility model (red curves) is, for all the magnitudes, leads to larger values in comparison with the values obtained
    from the analytically derived fragility functions (purple). These differences increase with magnitude. This feature
    might arise, not only from the fact that empirical fragility functions consider both earthquake and tsunami actions
    while the purple curves consider only the effects of the tsunami, but also because empirical fragility functions only
    account for flow depth as the IM. Conversely, the analytical fragility functions implemented were derived using the
    theoretical forces associated with the flow velocity tsunami waves as input in the generating numerical model. Similar
    observations regarding the reduction in the loss estimations when flow velocity is included have been drawn by other
    studies (e.g., Attary et al., 2019; Park et al., 2017).

2.  We observe that the ground-shaking dominates the losses at lower magnitudes (Mw 8.5, 8.6), whilst the tsunami,
    either from analytical (emulating far-field tsunamis) or empirical fragility functions (near-field tsunamis), controls
    the losses for the rest of the scenarios with larger magnitudes. The former is aligned with the observations of Goda





and De Risi, (2018) and Gómez Zapata et al., (2021e) for the case of empirical tsunami models. Moreover, a similar trend is observed for the disaggregated tsunami-induced losses (assuming initial non-zero damage) whose respective loss values (orange curves) are larger than the shaking-induced losses for M$_w$ 8.8, 8.9, and 9.0. Hence, these features highlight that as the magnitude increases, there is an increasing comparative importance of the tsunami risk within
the considered sequence of hazards.

3.    Expected loss values from cumulative damages based on single-hazard vulnerability models (our method, green curves) are clearly different from the one produced by classical empirical tsunami models. Classical empirical tsunami fragility functions lead to considerable lower losses estimations for the low magnitudes earthquakes and substantial larger estimations for the larger ones.

4.    The differences between the loss exceedance curves derived from both sets of analytical fragility models (either from undamaged or with pre-existing damage) are larger for the lower magnitudes (M$_w$ 8.5, 8.6) and decrease with increasing magnitude. As the magnitude increases, there is an increasing tendency of convergence between these two loss curves (M$_w$ 8.9, 9.0).

5.    Consequently, since tsunami-induced losses either from analytical fragilities (initial undamaged states) or from state-
dependent and inter-scheme models converge for the larger magnitudes (M$_w$ 8.9, 9.0), their respective summations with the shaking-induced losses would be approximately similar at the largest probabilities of exceedance. Nevertheless, this observation needs to be better investigated through more exhaustive simulations of tsunami inundation per considered scenario.

6.    Conversely, considering observation 3, (i.e., as the magnitude decreases, the differences between purple curves and
orange curves increases), their respective summations with the shaking-induced losses will lead to very different results. Hence, this observation suggests that, although earthquake and tsunami structural responses can be separately approximated for very large magnitudes, it is still required to address cumulative damages from the vulnerability interactions that are expected on the lower magnitudes earthquakes we have considered (i.e., M$_w$. 8.5, 8.6).

When we consider analytical fragility functions with $D_{k0}^A$ that only emulate the damaging actions of far-field tsunamis
(without any ground shaking), we observe that as the magnitude increases, their respective loss exceedance curves converge with the ones that assumed state-dependency ($D_{kz}^A$). This is because, for the larger magnitude events, the damaging actions due to seismic ground shaking will correspondingly increase. Hence, the available probabilistic damage transitions from the damage states within the earthquake (source) to tsunami (target) schemes will be consequently reduced. Therefore, we observe that if far-field analytical tsunami fragility functions are used, their corresponding results will be very much alike, regardless
of whether they are considered as being undamaged ($D_{k0}^A$) or with pre-existing damage ($D_{kz}^A$). Therefore, for these larger magnitude events, regardless of which curve is summed up with the shaking-induced losses, the resulting loss distributions for the hazard sequence would lead to quite similar results. Thus, the implementation of state-dependency on tsunami fragility may not be fully necessary to be addressed for very large earthquake magnitudes (M$_w$ 8.9, 9.0). This observation is aligned





with studies (i.e., (Petrone et al., 2020; Rossetto et al., 2018)) that suggest that earthquake and tsunami structural responses
can be separately approximated. However, the former statement would not apply to the low magnitude earthquakes investigated
in Lima for which the pre-existing damage due to earthquakes must be addressed. No generalizations should therefore be done
in this regard, with sensitivity analyses needing to be carried out in the future.

## 5 Discussion

    This study has proposed a modular method to disaggregate the direct losses expected for building portfolios exposed
to consecutive hazardous scenarios of different natures in which their individual components could be individually improved.
Therefore, future sensitivity analyses on some of the modules related to damage-state would benefit the understanding of how
their embedded uncertainties would impact their corresponding results. We can mention:

1. The disaggregation of building classes into taxonomic attributes as presented in Sect. 2.1 is an important input to
obtain the probabilistic inter-scheme building compatibility matrices based on Gómez Zapata et al., (2022b).
However, it is worth noting the shortcoming described by Charvet et al., (2017) referring to the generalised poor
taxonomic building characterizations of the currently available tsunami fragility models. They are, most of the time,
only based on their main construction material, although sometimes they include the number of storeys, and rarely
do they include other attributes such as the date of construction (e.g., Suppasri et al., (2015). When more enriched
descriptions for tsunami vulnerability get available in the future, this approach will remain useful for similar
purposes.

2. When/if local high quality empirical data collection and analytical models), become available, they could be used to
constrain the relationships between the failure mechanisms and attribute relevance for hazard-related susceptibilities.
This might contribute to enhance the construction of heuristics that characterise the likely observable damage extent
(per damage limit state, building type and hazard-dependent fragility model), that could be obtained through more
refined approaches such as unsupervised machine learning. Its use applied on real datasets that document
observations on building components (even different from the ones presented in Eq. 3) could contribute to refine
state-dependent tsunami fragility functions and to restrict the heuristics on the likely observable damage (Sect. 2.2)
and thus, minimizing subjective expert judgment. In this sense, it is worth noting that the set of predicted likelihood
probabilities in the probabilistic compatibility degree between damage states from different hazard fragility functions
that we derived from the synthetic datasets created through the heuristics and the AeDES scoring system are not
unique, as they depend on the choice of machine learning technique and on the heuristics derived through expert
elicitation. In this sense, we have documented a preliminary sensitivity analysis on such parametrization in Gómez
Zapata et al., (2022c). However, further investigation of the impact of such parametrization is still advised.



3.  As described by Hill and Rossetto, (2008), we have observed that, when characterising damage states due to the impacts of natural hazards on buildings, there is still the need for standardisation in describing observable physical damage after any kind of hazardous event through the harmonisation of damage scales for data collection, not only on entire building units but also regarding the particular damage (and extent) experienced by certain individual components. In this regard, although we have used the AeDES scale, other damage scales could be more suitable to describe the observable damage to some building classes than for others (Hill and Rossetto, 2008; Turchi et al., 2022). Nonetheless, the choice of a standard scale to transversally describe any observable set of damage on buildings will benefit the research in multi-hazard vulnerabilities.

4.  The integration of economic consequence models for physical vulnerability based on the replacement costs as a function of the buildings' area, as for instance presented in Triantafyllou et al., (2019) for tsunami vulnerability is worth testing. This also depend on the available data and it is out of the scope of this paper, but it would be worth exploring their contribution once more refined estimations about replacement cost are available for Lima. Nevertheless, one should be aware on the uncertainties involved for large scale building exposure models.

The derivation of the hazard intensities could also benefit from future enhancements. For instance, the GMPE-based seismic accelerations derived on a simplified $Vs_{30}$ site-grid of ~1 km might be too coarse to capture local site effects in the expected ground motions. However, the performance of site-response analyses that account for the local geotechnical soil properties of site-specific soil profiles, as for instance reported by Aguilar et al., (2019), is a computationally demanding task that is out of the scope of this study, but when integrated it could benefit the overall quality of seismic risk calculations for the study area. Complementary, we strongly advise the physical to generate exhaustive sets of cross-correlated-ground motion fields (at the required spectral periods by the buildings classes) to address their aleatory uncertainty. The selection of this model, among the available ones, carries epistemic uncertainties.

It is worth noting that the variability of the loss exceedance curves obtained for the cumulative damage (due to tsunamis) was derived from the damaged exposure models subjected to each realisation of cross-correlated ground motion fields (i.e., orange curves in Figure 12). Therefore, investigating the impact of other tsunami vulnerability and hazard data products (Behrens et al., 2021), which was beyond the scope of this paper, are nonetheless worth exploring. When such parametrisation in the tsunami data products becomes available for Lima, future studies could provide dimensionality of the contribution of the tsunami hazard upon the outlined method for scenario multi-risk estimates.

For the commonly exposed residential building stock of Lima exposed to both perils, we have observed that assuming initial undamaged states in the selected tsunami empirical fragility functions leads to large underestimations for lower magnitudes ($M_w$) and large overestimations for larger $M_w$ events in comparison to when state-dependent models were used. Hence, the initial "undamaged state" assumption used to assess the tsunami vulnerability in former studies (e.g., Adriano et al., 2014; Gómez Zapata et al., 2021e) may not be completely accurate to represent the losses expected after this type of cascading sequence. This is because such an assumption misses the calculation of earthquake-related damage which is an





important input needed to assess cumulative damage and losses through state-dependent analytical fragility models. On the other hand, adopting the larger value between independent earthquake and tsunami risk computations proposed by Goda and De Risi, (2018) may lead to better correspondence with our model (mostly for the lower $M_w$ events) than the sole use of the selected non-state dependent analytical fragility functions.

## 6    Conclusions

We have proposed a modular method that allows us to consistently re-use existing single hazard fragility models that are being developed by experts in various research fields and integrate each other for multi-hazard risk assessment for extended building portfolios. This integration aims for the probabilistic harmonisation of diverse hazard-dependent building classes and damage states which are included in their associated fragility functions. Through this integration, we aim to provide an alternative approach to conventional ones (e.g., HAZUS-MH (FEMA, 2003, 2017)) that consider a single building class with sets of fragility functions for a variety of hazards. In this sense, the method we have developed can be particularly useful to assess the cumulative damage in hazard sequences of different natures and forces that might induce various failure mechanisms upon the exposed buildings. Thereby, the presented integrative method contributes to reducing the existing gaps due to the typical lack of collective calibration and validation of multi-hazard risk methods. This is due, for instance, when triggered events act on damaged assets right after the first hazard or even simultaneously experiencing compound hazards with no time for damage reconnaissance or disaggregation of the damage features induced by the individual hazards.

We have proposed a modular method composed of the following components:

1.  The selection of existing hazard-dependent vulnerability schemes to model the building portfolio under each hazard-dependent vulnerability scheme of interest. They contain sets of building classes and associated fragility functions. To model the physical vulnerability of the building portfolio towards the triggering event (in this case, earthquake), no preference on whether empirical or analytical fragility functions should be used.
2.  On the other hand, to model the physical vulnerability of the building stock towards the triggered event, sets of state-dependent fragility functions must be derived for each building type within the selected scheme. For this purpose, it is important to use models that do not involve the damaging effects of the triggered event as the starting point. (i.e., avoiding empirical models and using analytical ones). This proposal overcomes the assumption of initial undamaged states for the structures exposed to the triggered event and allows to account for the differential cumulative damage between hazards.
3.  The characterisation of building classes through their disaggregation into building taxonomic attributes. This description allows the harmonisation between the building classes belonging to different hazard-dependent vulnerability schemes through the probabilistic inter-scheme compatibility matrix proposed in Gómez Zapata et al., (2022b).





4. The exposure models are spatially aggregated into optimal geographic entities (i.e. CVT-based models) that account for the spatial variability of low-correlated hazard IM in their derivation (Gómez Zapata et al., 2021e). This selection

was taken due to performance purposes only, but a more refined block-based model could also have been used.

5. A generalized description of the damage states based on a set of observable damage types on individual building components. This is done through a scoring system based on an underlying common scale (employing, for example, the AeDES form) that ultimately allows us to get the damage-state inter-scheme conversion. We use the total probability theorem, a Bayesian formulation, and machine learning techniques.

6. The vulnerability assessment for sequences of cascading hazards scenarios through the proposal of consistent economical consequence models across hazard-dependent vulnerability schemes. They must define replacement cost ratios per damage state and per fragility function associated with each vulnerability scheme.

The joint combination of these components creates a method to update the damage states throughout the multi-hazard sequence while allowing us to exploit existing hazard-specific risk-oriented taxonomies (i.e., building classifications with

corresponding fragility functions and defined damage states) available in the literature for a wide range of natural hazards. This is a modular method in which each one of their individual components can be separately customized when seeking future improvements.

When applying this method on the residential building stock of Lima (Peru), we have observed, on the one hand, that considering the risk metrics from tsunami vulnerability only from the selected set of empirical fragility functions (derived from

near-field tsunamis) as representative of the shaking and tsunami sequences leads to underestimations for the lower magnitudes. On the other hand, we have observed overestimations for the larger magnitude scenarios in comparison with the state-dependent method that accounts for the accumulated damage due to the former earthquake solicitations. We have observed that the use of the proposed method to assess the cumulative damage is more relevant for the lower magnitude scenarios than we have considered ($M_w$ 8.5 and 8.6). This might be due to the greater damage extension on the exposed

buildings that is expected from the seismic demands in comparison with those imposed by their corresponding tsunamis, and thus, there is greater chance to obtain cumulative damage. On the contrary, for larger magnitudes, the use of state-dependent fragilities and analytical functions assuming no pre-existing damage are converging, and thus, the importance of assessing state-dependency is reduced.

Considering the limitations and simplifications assumed in this study, we are not claiming that the resulting economic

losses we have calculated for the residential building stock of Lima from multi-hazard scenario-based risk computations are totally exhaustive. Thus, caution should be taken with the interpretation and extrapolation of these conclusions to other study areas and combinations of models. Nevertheless, awareness of these uncertainties for the reliable quantification of risk towards these cascading hazards is increasingly important to enhance mitigation strategies for disaster risk reduction (Imamura et al., 2019). Furthermore, it is worth recalling that the method herein proposed has been exclusively designed for spatially extended

residential building buildings as a proof of concept for integrating existing fragility models. We do not provide an\ complete



validation of multi-vulnerabilities approaches, but rather we offer a holistic and novel harmonising method to track such dynamics in a consistent manner. Hence, our method is not meant to replace more detailed analytical analyses required to determine the structural response of individual buildings subjected to seismic and tsunami loading (e.g., Petrone et al., 2017; Rossetto et al., 2019).


***Code and data availability***. The data used in the elaboration of this study are available in open repositories. The scenario-based ground motions and tsunami inundation maps are available in Gómez Zapata et al., (2021c); Harig and Rakowsky, (2021), respectively. The first set was calculated making use of the Shakyground script (Weatherill et al., 2021) which relies on the OpenQuake Engine (Pagani et al., 2014), whilst the second set was calculated using the TsunAWI software. The exposure and fragility models for both hazard-vulnerability schemes (earthquake and tsunami) are available in Gómez Zapata et al., (2021a, b) and were adapted to fulfil the data formats required by the scripts provided by Assetmaster and Modelprop (Pittore et al., 2021). They were used as inputs for the scenario-based seismic risk assessment (Sect. 3.6) using the DEUS software (Brinckmann et al., 2021). The scenario-based risk estimates for earthquakes and tsunami using analytical and empirical fragility functions respectively are provided in Gómez Zapata et al., (2021d). State-dependent analytical tsunami fragility functions used in this study are available in Gómez Zapata et al., (2022a). The set of inter-scheme damage compatibility matrices used in this study are provided in Gómez Zapata et al., (2022c).

***Competing interests***. The authors declare that they have no conflict of interest. The funders had no role in the design of the study; in the collection, analysis, or interpretation of the data; in the writing of the manuscript; or in the decision to publish the results.

***Funding***. This research was funded by the RIESGOS and RIESGOS 2.0 projects, funded by the German Federal Ministry of Education and Research (BMBF), with Grant No. 03G0876A-J and 03G0905A-H, respectively. These projects are part of the funding programme CLIENT II – International Partnerships for Sustainable Innovations'.

***Acknowledgments.*** The authors want to express their gratitude to Andrey Babeyko, Michael Haas, Michael Langbein, Giuseppe Nicodemo, Cecilia Nievas, Juan Páez-Ramírez, Juan Palomino, Matthias Rüster, and Sandra Santa-Cruz for their support during the elaboration of this study. Thanks to Sven Harig and Natalja Rakowsky for having provided us with the tsunami inundation models for Lima. We also thank Henning Lilienkamp and Graeme Weatherill for their support with the simulation of spatially correlated ground motion fields and machine learning techniques. We thank Kevin Fleming for the careful proofreading.

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
