# Peer review of "Scenario-based multi-risk assessment from existing single-hazard vulnerability models. An application to consecutive earthquakes and tsunamis in Lima, Peru"

_Natural Hazards and Earth System Sciences, 2022_

## Author Comment (AC1)

**Responses to Anonymous Reviewer 1.**

Thank you for your helpful review. Please find our answers to each of your comments below

1. >> *"The paper properly describes the topic declared by the authors. The different parts of the proposed method are presented in a detailed way, with a rich literature reference. The topic of the paper affords a challenge in the field of multirisk loss assessment, so, the comments presented in the discussion (limits and positive aspects) are agreeable".*

    Thank you for your nice comments and your suggestions.

*In addition:*

2. >> *"A re-reading of the paper is suggested to correct some typing errors and just some language errors;"*

    Thank you for the nice comment about the structure of the paper. Following your advice, we have accordingly asked an editor (a native English speaker) to carry out a strict language review. The new version has been significantly been improved in that regard.

3. >> *"In the last paragraphs check the use of the numbered list, to extend it to the final sentences;"*

    We have performed such a check in the updated version of the manuscript. We agree with the reviewer that the paragraph after point 4 of the Discussion section better fits as a point 5 because it also discuss the limitations we faced regarding hazard intensities. Thank you for your suggestion.

4. >> *"A check of the conclusion and discussion paragraph is suggested to avoid some repetitions."*

    We have performed such a check in the updated version of the manuscript. We believe that the overall points we have addressed in these sections are now better presented. Thank you.

    Complementarily to the reviewer's suggestions, we have also made slight modifications to the Introduction and Sect. 2.3 that will provide the reader with a smoother reading. Moreover, we have included a simple but informative analysis comparing our results with other scientific literature results at the end of the Discussion section:

    *"To give a perspective on the importance of addressing cumulative damage and losses for building stocks, let us recall some of* the findings that the available studies of Gómez Zapata et al., (2021) and Markhvida et al., (2017) found. They investigated the likely economic losses of the entire residential building portfolio Lima and Callao solely after seismic ground motion from a $M_w$ 8.8 scenario addressing the variability induced by the same cross-correlation model we have implemented herein. In the first *study, ~1,657,635 residential buildings were considered and both studies considered the SARA building classes and fragility functions, similar to what we have done. Both studies reported mean loss values of around 7 and a maximum of around USD 35 billion (among a stochastic sample of events). It is then interesting to compare such a range of values with the mean loss values reported for a similar $M_w$ (Fig 11-d). Notably, the forecasted losses per event (shaking and tsunami) and inferred from cumulative damage were derived from the much smaller commonly exposed building stock to*

*each pair of hazard scenarios (see Fig 8-c), which constitute ~ 21,209 buildings. This means that the building count for the entire residential stock of Lima (Fig 6-a) is around 78 times larger than the commonly exposed to both perils (Fig 6-b). Hence, can note the important role of tsunami-induced losses in the study area. The mean losses expected from the cascading sequence of that $M_w$ 8.8 (i.e. value for the $50^{th}$ percentile on the green curve in Fig. 11b) is ~ USD 0.75 billion and a maximum of around USD 0.94 billion. Therefore, given the difference between the size of both building portfolios, finding out that the losses for the entire city are expected to be only 9 times larger than the ones forecasted after the action of both earthquake and tsunami, tells us that the crucial importance of carefully addressing the cumulative damage due to tsunami in the study area. Moreover, this tell us that, besides all of the secondary effects of the tsunami, these types of future scenarios in Lima will constitute a huge driving source of direct economic losses for building portfolios, but also uncertainties due to the lack of data to calibrate or validate these types of risk assessment after the action of cascading hazards".*

**References**

Gómez Zapata, J. C., Brinckmann, N., Harig, S., Zafrir, R., Pittore, M., Cotton, F., and Babeyko, A.: Variable-resolution building exposure modelling for earthquake and tsunami scenario-based risk assessment. An application case in Lima, Peru, Natural Hazards and Earth System Sciences, 21, 3599–3628, https://doi.org/10.5194/nhess-21-3599-2021, 2021.

Markhvida, M., Ceferino, L., and Baker, J. W.: Effect of ground motion correlation on regional seismic lossestimation: application to Lima, Peru using across-correlated principal component analysis model, Safety, Reliability, Risk, Resilience and Sustainability of Structures and Infrastructure. 12th Int. Conf. on Structural Safety and Reliability, Vienna, Austria, 2017.

---

## Author Comment (AC2)

**Responses to Anonymous Reviewer 2.**

Thank you for your helpful review. Please find our answers to each of your comments below.

**A. General comments**

>>"*The manuscript investigates a novel method for accounting for damage accumulation on large building portfolios exposed to sequential earthquake and tsunami hazards. This work is of great interest to the readers of NHESS. The paper is well-structured and clear, despite very minor typos. However, some contents of the current manuscript need to be clarified and a few sections to be improved before the paper can be considered for publication*".

Thank you for your nice comments about the structure of the paper and your suggestions. Following your advice, we have accordingly asked an editor (a native English speaker) to carry out a strict language review. The new version has been significantly been improved in that regard.

1. >> "*A.1 Lines 100-103. The purpose of this study is stated in these three sentences. The aim is relatively clear, however it remains unclear how the state-dependent fragility models are obtained. This is one of the critical parts of this methodology and should be briefly explained at this stage, as this would help the readers*".

   Thank you. We really appreciate your constructive comment. We agree that we had formerly mentioned the details about the derivation of the tsunami state-dependent fragility functions at a late stage. For instance, the citation to the open data repository Gómez Zapata et al., (2022a) was cited in line 775 despite the fact that the dataset was mentioned for the first time on line 325 where details about the specific selected model were provided (Sect. 3.3). In the updated version, the reference has been included at the end of Sect. 2.3.

   More details have also been provided and the following lines have been added:

   "*For multi-risk assessment, such a damage-updated exposure model (i.e. building classes and damage limit states) requires to be associated with a set of tsunami fragility functions that can account for the pre-existing damage caused by the first event (shaking). In the application chapter of this paper, a set of tsunami state-dependent fragility functions are obtained through the use of simple ad-hoc scaling factors. Nonetheless, it is important to stress that these functions should be replaced by state-dependent tsunami fragility functions derived from more sophisticated methods as soon as they would become available*".

   We sincerely think that these new lines will provide a smother reading experience. Thank you.

2. >> "*A.2 Lines 125-130. Regarding the integration of a set of modular components, the authors should provide a clearer definition of "inter-scheme compatibilities" and "their related compatibility levels between inter-scheme damage states". I recognise that these concepts are described in more detail in the following sections, but this paragraph is quite difficult to read. It is not clear what these 'conversions' are… *".

   We fully agree with the reviewer's comment. Although the method to derive the first set of these inter-scheme compatibilities i.e., between 2 sets of building classes, summarised in Sect.

2.1, was already formally presented in a separate publication Gómez Zapata et al., (2022b), we have now realized that the motivations to use such an approach were not clearly enough stated. Therefore we have included the following sentence:

*"By reusing this approach, a building stock initially classified into earthquake-oriented typologies can now be probabilistically represented by tsunami-oriented typologies".*

Furthermore, more clarity can be achieved if we state that these "conversions" are ultimately represented through two sets of *"compatibility matrices"*, which are probabilistically generated by the methods explained in detail afterward. We have also realized that the word "matrix" is clue herein and it was missing. We have therefore modified the sentence as follows:

*"The two aforementioned conversions are ultimately represented through two sets of "compatibility matrices" that are probabilistically generated. The advantage of using these matrices is that through these conversions, the damaged updated exposure model resulting from the action of $IM^A$ can be represented in the domain of the reference scheme attached to the second vulnerability to be analysed, and the damage updated exposure model can be directly used for a second risk computation, in this case for tsunami risk addressing the cumulative damage".*

**Reviewers continues** >> *"…and how they make it possible for a cumulative assessment of the damage".*

In order to provide clarity in this regard, even at this early stage of the paper, we have decided to complement the former part of the same paragraph by adding the following explanation.

*"This is achieved by using **(3)** generic state-dependent tsunami fragility functions (i.e. with non-zero initial damage states made of new curves that represent the permissible damage progression). Since the resultant earthquake-induced damages are formerly expressed in the tsunami vulnerability domain (step 2), the non-zero damage limit states of this set of state-dependent tsunami fragility functions will implicitly account for such pre-existing damage. The joint ensemble of these three components can be ultimately used to calculate the cumulative expected damages after the triggered event with $IM^B$, while accounting for the preceding induced by $IM^A$ (i.e., blue part in Figure 1b, developed in Sect. 2.3).*

3. >> *"A.3 Lines 160-177. The AeDES forms have been consistently used for post-earthquake damage assessment. The authors state that these can also be used for tsunami: why? Also, did the expert elicitation help to adapt these forms to tsunami damage?"*

We understand the reviewer's concern about this issue. However, we should emphasize some ideas that perhaps were not presented clearly enough.

One of the main current problems faced by scientists working on the derivation of empirical fragility functions for assets exposed to various hazards (i.e. that require damage reconnaissance campaigns) is the absence of a standard scale to assess the observable damage in a comparable manner across hazard-dependent vulnerabilities. The reviewer is

fully right that the AeDES scale (Baggio et al., 2007) had never been used to assess tsunami-induced damage before.We have considered that that aspect is not problematic, but it is rather an opportunity to demonstrate that the observable physical damage incurred by any kind of hazard can be transversally described by using "low-level taxonomic elements" (i.e. a set of individual building components).

The proposal of making use of the AeDES scale in the application section of the paper is not determinant nor compulsory for the usability of the proposed method. Actually, in point 3 of the discussion section (lines 553-556) we had already stated *"In this regard, although we have used the AeDES scale, other damage scales could be more suitable to describe the observable damage to some building classes than for others"*.

For instance, the EMS-98 scale (Grünthal, 1998) with 5 categories originally proposed for ground-shaking damage, had been the selected damage scale used to assess the damage extent on building types at the beginning of our study through expert elicitation. Notably, as we described in lines 53-55, the EMS-98 vulnerability classes have the great advantage that it has been already used not only to assess the likely observable damage due to seismic action, but it has been also used to classify likely ranges of vulnerabilities to other hazards (i.e. tsunami, floods, strong winds) based on the building's material types (Schwarz et al., 2019; Maiwald and Schwarz, 2019). However, since the scope of implementing a standard damage scale was ultimately to generate a synthetic dataset of observable damages for the various building types, the 0-5 scale of the EMS-98 was too limited. In fact, we noticed that using this scale, many of the damage features on some building components overlapped across damage limit states (i.e. they were not clearly distinguishable between adjacent damage states, e.g. infills and partitions for slight and moderate damage).

The selection of the EMS-98 damage scale resulted in a very limited number of produced data points in this synthetic collection, and hence not exhaustive enough to produce comparatively more robust probabilistic inter-scheme damage compatibility matrices. On the contrary, having used the AeDES scale, with a 0-9 scale and already related to the four selected specific building components (as we did in the submitted version of our paper), allowed us to generate a more exhaustive synthetic dataset (see Fig. 8 of the submitted version), and thus, we consider that this section also allowed us to produce better quality damage-related conversion matrices (Fig. 9 of the submitted version).

Details about the expert elicitation used in the AeDES form for tsunami-induced damage will be described in the answer to the next comment.

4. >> *"A.4 Lines 178-185. What are the details of the expert elicitation? Who took part in this exercise? Did the expert elicitation involve experts in earthquake and tsunami engineering?"*

The expert elicitation for the two hazard vulnerabilities of our interest, as explained in the paper, consists in the selection of the damage level (among the 0-9) scale that is likely to be observed for each damage limit state and building typology. In the submitted and present version we cite the open data repository Gómez Zapata et al., (2022c) which contains a *Jupyter notebook* and *Python scripts* where these scores and the subsequent process of using the derived synthetic dataset to classify their respective compatibility through machine learning.

Expert elicitation for earthquake vulnerability was supported by Dr. Nicola Tarque, co-author of our paper. He is also a co-author of the Villar-Vega et al., (2017) study that proposed the seismic fragility functions for the SARA building classes, upon we have based our calculations. Nicola Tarque is originally a Peruvian researcher, with extensive experience in deriving fragility functions for masonry, adobe and reinforced concrete buildings in Peru[1], and hence, he is aware of the damage features that can be expected from the damage limit states defined within this set of fragility functions for the local Peruvian building types contained in the exposure model we adopted.

On the other hand, considering that the building typologies contained in the Medina scheme have analytical tsunami fragility functions associated, the AeDES-based scores used to generate the resultant synthetic dataset were constrained through expert elicitation by thinking as if the tsunami forces alone were the cause of the described damage features (i.e. far-field tsunami). For this exercise, the authors of the models proposed by Medina et al., (2019), who are also co-authors of our paper (Dr. Juan Lizarazo and Sergio Medina) contributed to selecting the observable damage features for the four damage limit states of every fragility function associated with the selected set of six building types that they had defined in their previous study Medina, (2019). They are two recognized members of the tsunami-vulnerability community, who for instance have had numerous research exchanges in the framework of the Japanese-Colombian project SATREPS[2] about multi-hazard risk.

***Reviewers continues*** >> *"What are Scheme A and Scheme B? Are A and B earthquake and tsunami, respectively? It is quite confusing"*

Scheme *A* and Scheme *B* are referring to the selected models for earthquake and tsunami vulnerability (i.e. in the application section: SARA and Medina, respectively). Such a definition had been presented in Figure 1-a as well as in several parts of the submitted version of our manuscript.

5. >> *"A.5 Lines 203-227. State-dependent fragility functions are developed for accounting for the cumulative damage. It is not clear the fragility function for a building type damaged by the earthquake and then by the tsunami is obtained. What are the scaling factors? How are these calculated, based on what ad-hoc calibration? The authors should clarify this key element, which remains quite obscure in the application as well".*

We deeply thank the reviewer for his/her comment.

It should be noted that we used and cited the open data repository Gómez Zapata et al., (2022a) where the assumptions to derive such ad-hoc state-dependent tsunami fragility functions are documented. This repository was and is also cited in the "Code and data availability" section (line 659) and it is accessible through the URL provided (line 775 of the originally submitted version). It is worth noting that this data repository has currently a temporary review link, and its designated DOI will be available once this paper has been published.
* * *
[1] https://www.researchgate.net/profile/Nicola-Tarque-2/research
[2] http://agenciadenoticias.unal.edu.co/detalle/unal-prepares-model-to-measure-tsunami-impact

Therefore, we have done our best to ensure that involved assumptions and software are well documented and avoid the "obscurity" of "black boxes". Making available this information through the GFZ data service is a transparent practice that our team has been pushing for some time. Hence, the readers will be able to compare and possibly improve our approaches in the future.

However, the reviewer's comment has helped us to realise that some of the related ideas regarding these topics were not clearly expressed within the text. Since the application part of the method (Sect. 3.3) fully depends on the selected set of fragility functions that are linked to the building classes for tsunami vulnerability (i.e., Medina, 2019), no generalization was done in Sect. 2.3. However, as already stated in the response to comment "A.1", the open data repository open data repository (Gómez Zapata et al., 2022a) has been now cited at the end of Sect. 2.3 of the updated version. Then, the reader is able since an earlier stage to get more details about the derivation of the ad-hoc scaling factors.

It is worth noting that a part of Sect. 2.3 was formerly devoted to present the assumptions of our approach, which is largely based on the ideas of Mignan et al., (2014) (please see page 4, Fig. 6). Considering the former citation as well as the pertinence of having cited the auxiliary data repository, we have carefully evaluated the necessity of repeating some of the assumptions that the aforementioned paper already documents. Therefore, one of the main modification that we have made to the revised manuscript is having removed Equation 9. This is because this equation can be consulted in the aforementioned reference, it is simply a generic expression of a lognormal cumulative distribution and as the reviewer correctly noticed it was not sufficiently pointing out the assumptions we followed to derive state-dependent fragility functions. Hence, we have now included some of the theoretical assumptions we that we followed in the updated version of the paper, which have been always documented in the cited data repository (Gómez Zapata et al., 2022a), where the reader can find the specific example case we tested in Sect. 3.3. The following has been included:

*Only for the overall scope of this paper, we propose that state-dependent fragility functions can be simplified by using ad-hoc calibration parameters to modify these logarithmic mean values. For such a modification, we propose applying to them the exponential operator to obtain the physically accountable mean IM (hazard intensity measures). I.e., $\lambda_{q_0}(T_r^I)$ defines each damage state as: $\lambda_{q_0}(T_r^i) = e^{\mu_{q_0}(T_r^i)}$. Subsequently we propose to obtain their respective differences $\Delta\lambda_{q_0}$. For example, if a fragility function is composed of $q_{N_i} = 4$ damage states (excluding damage state 0, equivalent to no damage), there will be a set of damage states $\lambda_{q_0} = [\lambda_{q_{0_1}}, \lambda_{q_{0_2}}, \lambda_{q_{0_3}}, \dots \lambda_{q_{0_i}}]$ for which we should obtain the differences between all the possible top and bottom damage states and we must obtain six values: $\Delta\lambda_{q_0} = [\Delta\lambda_{q_{0_{1,2}}}, \Delta\lambda_{q_{0_{1,3}}}, \Delta\lambda_{q_{0_{1,4}}}, \Delta\lambda_{q_{0_{2,3}}}, \Delta\lambda_{q_{0_{2,4}}}, \Delta\lambda_{q_{0_{3,4}}}]$. In this example, these six state-dependent transition values are included within the $G_f = 10$ triangular number (i.e 4 from 0; 3 from 1; 2 from 2; 1 from 3) given by **Error! Reference source not found.** Thereby, for each $T_r^i$, it is still necessary to determine the probabilistic representation (log mean and log standard deviation) of every damage state transitions $\Delta\lambda_{q_0}$. To do so, the $\lambda_{q_0}(T_r^i)$ values are proposed to be multiplied by the $\Delta\lambda_{q_0}$ factors, and reframing this quantity to a natural logarithm in order to approximate it back again to lognormal mean values. This is expressed as given by Eq. 1.*

$$\delta_{w|y} = ln(\Delta\lambda_{q_0} \times \lambda_{q_0}).$$  *Eq. 1*

*The reader should note that in this approach, the $\Delta\lambda_{q_0}$ vales are a set of ad-hoc calibration parameters or scaling factors that are applied directly to the IM $\lambda_{q_0}$ for which each damage limit state was originally*

*derived. These values form the lognormal mean of the state-dependent fragility functions. A similar approach was followed by Rao et al., (2017). The fragility functions used to constrain the state-dependent fragility functions should have been derived only for the actual second acting hazard (i.e., far-field tsunamis). Thus, the use of those derived analytically is advised over empirical ones (which had implicit the damaged induced by ground-shaking in their derivation). Further details about this approach and model assumptions to find the ad-hoc calibration parameters are provided for the example case in the data repository Gómez Zapata et al., (2022a).*

The methodological aspect of the former assumption is of course open for improvement, but as we stated in the discussion section (lines 540-542): the acquisition of "*real datasets that document observations on building components (even different from the ones presented in Eq. 3) could contribute to refine state-dependent tsunami fragility functions (…)*". Hence, the ad-hoc scaling factors are only place-holders within the method we propose and other, more sophisticated methods should replace this simplistic, but pragmatic working hypothesis.

6. >> *"A.6 Figure 11. I would present a similar work-flow for the general methodology earlier in the manuscript".*

   We sincerely thought that figure 1-b was informative enough to present some basic concepts, and that a workflow only for the actual application case will only make sense once the punctual study area and the enclosed assumptions (as shown in Fig. 11) are first clearly presented. Another figure could be seen as redundant since the new figure would contain the sentences that are already presented in Fig. 11 (without the visual advantage of showing a geographically located earthquake rupture, exposure model, damage forecast, fragility functions). We do believe that the graphics within this figure are really helpful to the reader to understand the overall method. To take into account the reviewer comment we suggest that this figure can be the "selected main figure" of our paper. This is a great alternative that the NHESS journal offers to show the "most important figure of the manuscript" (please see **here**[3] how Fig 5 of this paper is shown as a "visual abstract" directly on the main web version of the paper).

**B. Specific comments**

1. B.1 Line 34: Indian Ocean Tsunami:  *Corrected.*

2. B.2 Line 83: consider also Petrone et al., (2017) a relevant study on tsunami analytical fragility functions for a single building.
   *Ok. It has been added as suggested. Thank you. However, please note that this reference was already provided somewhere in the text for a similar context (line 649) to not be repetitive.*

3. B.3 Line 96: "to" repeated twice: *Corrected*.

4. B.4 Line 111: "scenario" instead of "scenarios": *Corrected*.

5. B.5 Line 114. Please clarify the meaning of "vulnerability modes"
* * *
[3] https://nhess.copernicus.org/articles/21/3599/2021/

*We meant "vulnerability models". Sorry about it. It has been corrected.*

6.  B.6 Line 127. The authors refer to the "purple part" and then "red part" and so on – however the use of colours might be challenging if the paper is printed using greyscale. I would suggest to use a different way to identify the different components in Figure 1b.

    *The use of colours in all of the figures is of our preference. The paper will be print in colour. We have experience publishing in the NHESS journal and we can ensure you that it will be that way. You should not worry about it. Thanks anyways.*

7.  B.7 Line 645. Amend typo "an\": Corrected.

**References**

Baggio, C., Bernardini, A., Colozza, R., Corazza, L., Della Orsini, M., Di Pascuale, G., Dolce, M., Goretti, A., Martinelli, A., Orsini, G., Papa, F., and Zuccaro, G.: Field Manual for post-earthquake damage and safety assessment and short term countermeasures (AeDES), EUR 22868 EN – Joint Research Centre – Institute for the Protection and Security of the Citizen., : Office for Official Publications of the European Communities, Luxembourg, 100 pp. – 21.00 x 29.70 cm pp., 2007.

Gómez Zapata, J. C., Medina, S., and Lizarazo-Marriaga, J.: Creation of simplified state-dependent fragility functions through ad-hoc scaling factors to account for previous damage in a multi-hazard risk context. An application to flow-depth-based analytical tsunami fragility functions for the Pacific coast of South America, GFZ Data Services, 2022a.

Gómez Zapata, J. C., Pittore, M., Cotton, F., Lilienkamp, H., Simantini, S., Aguirre, P., and Hernan, S. M.: Epistemic uncertainty of probabilistic building exposure compositions in scenario-based earthquake loss models, Bulletin of Earthquake Engineering, https://doi.org/10.1007/s10518-021-01312-9, 2022b.

Gómez Zapata, J. C., Pittore, M., and Lizarazo, J. M.: Probabilistic inter-scheme compatibility matrices for multi-hazard exposure modeling. An application using existing vulnerability models for earthquakes and tsunami from synthetic datasets constructed using the AeDEs form through expert-based heuristics, GFZ Data Services, https://doi.org/10.5880/riesgos.2022.003, 2022c.

Grünthal, G.: European Macroseismic Scale 1998, Centre Européen  de Géodynamique et de Séismologie., Luxembourg, 1998.

Maiwald, H. and Schwarz, J.: Unified damage description and risk assessment of buildings under extreme natural hazards, Mauerwerk, 23, 95–111, https://doi.org/10.1002/dama.201910014, 2019.

Medina, S.: Zonificación de la vulnerabilidad física para edificaciones típicas en San Andrés de Tumaco, Costa Pacífica Colombiana, Master thesis in Civil Engineering, Universidad Nacional de Colombia  Facultad de Ingeniería, Departamento Ingeniería Civil y Ambiental, Bogotá, Colombia, 245 pp., 2019.

Medina, S., Lizarazo-Marriaga, J., Estrada, M., Koshimura, S., Mas, E., and Adriano, B.: Tsunami analytical fragility curves for the Colombian Pacific coast: A reinforced concrete building example, Engineering Structures, 196, 109309, https://doi.org/10.1016/j.engstruct.2019.109309, 2019.

Mignan, A., Wiemer, S., and Giardini, D.: The quantification of low-probability–high-consequences events: part I. A generic multi-risk approach, Natural Hazards, 73, 1999–2022, https://doi.org/10.1007/s11069-014-1178-4, 2014.

Petrone, C., Rossetto, T., and Goda, K.: Fragility assessment of a RC structure under tsunami actions via nonlinear static and dynamic analyses, Engineering Structures, 136, 36–53, https://doi.org/10.1016/j.engstruct.2017.01.013, 2017.

Rao, A. S., Lepech, M. D., and Kiremidjian, A.: Development of time-dependent fragility functions for deteriorating reinforced concrete bridge piers, null, 13, 67–83, https://doi.org/10.1080/15732479.2016.1198401, 2017.

Schwarz, J., Maiwald, H., Kaufmann, C., Langhammer, T., and Beinersdorf, S.: Conceptual basics and tools to assess the multi hazard vulnerability of existing buildings, Mauerwerk, 23, 246–264, https://doi.org/10.1002/dama.201910025, 2019.

Villar-Vega, M., Silva, V., Crowley, H., Yepes, C., Tarque, N., Acevedo, A. B., Hube, M. A., Gustavo, C. D., and María, H. S.: Development of a Fragility Model for the Residential Building Stock in South America, Earthquake Spectra, 33, 581–604, https://doi.org/10.1193/010716EQS005M, 2017.

---

## Author Response (AR2)

Response to the NHESS Editors,

Executive Editor, Prof. Dr. Bruce Malamud

Handling editor, Dr. Elisabeth Elisabeth Schoepfer

I would like to thank you for the positive response of accepting our manuscript pending minor corrections.

The technical corrections of the two anonymous reviewers have been satisfactorily solved in the revised version of the manuscript. Moreover, I would like to express my apologies for having submitted the PDF version with some figures with a wrong black background. That was a mistake that I did not noticed while uploading the file, but luckily the original Word version was undamaged. Thanks for your understanding.

I sincerely thank the helpful feedback of Prof. Dr. Bruce Malamud. Please find the answers to each of your comments. They have been listed using numbers (18 comments) in the case punctually addressing each suggestion is required in the future. With this, I consider that the manuscript has been significantly improved.

**1. Abstract.**

>>The Editor suggested to be more explicit with numbers and economic losses. However, he also said "**It is fine if you want to leave as is".** Therefore, we have decided to mostly leave it as it was written on the previous version of the manuscript. This is because we consider that the assumptions involved in the proposed method as well in the data used on the application part of the paper require a significant overview that cannot be given in the abstract, which to our opinion should tell the generalities of the paper while motivating to read other parts of the paper. However, the words "direct economic" (referring to the forecasted losses) has been added to this new version of the abstract to emphasize that only this type of metric was considered in our study.

**2. GENERAL.** This will be picked up in copy-editing, but throughout, remove ',' before (YYYY). So for example "Goda and De Risi (2018)" not ", (2018)". Park et al. (2019) not Park et al., (2019).

>>Before doing this change, I remembered that the 'al**.,**' is the advised style of citation by the NHESS journal. Please note that this is also included in the Zotero default style for NHESS. I also cross-checked a few of published papers in the same journal (including some authored by myself and by Prof. Dr. Bruce Malamud), and I could notice that the 'al**.,**' style was used in the final print version. I hope this explanation helps to understand why no change was done in this matter. In any case, if there is a recent formatting that I do not know well, I am sure that, the copy-editing service of the Journal will satisfactory fixed, as already written on this comment.

**3. INTRODUCTION.** The introduction meanders a bit in its text. It takes a while (paragraph 2) until the reader knows what exactly will be studied. Paragraph 2 does explain it, but you could be more explicit.

"This study narrows down the scope of **???cascading events, you decide on words????******** by assuming that a second hazardous event is always triggered after the occurrence of the first one," You want a self-contained sentence not dependent on the first paragraph.

>>This sentence has been updated as follows: *"Therefore, this study narrows down the scope of scenario-based multi-hazard risk by assuming that a second hazardous event is always triggered after the occurrence of the first one, thus eliminating the need to quantify the probability of this occurring".* Thank you for your helpful suggestion.

**4. INTRODUCTION.** For the second paragraph. At the end of it, tell us what will be in the rest of the introduction. "The rest of this introduction will discuss ***, ***, and ****" This will help the reader understand what is to come.

>>Following the Editor's helpful comment, we have introduced some new lines of text presented below. This will allow the reader having a smoother experience while reading the rest manuscript.

*'As a premise, this study contributes to the field by proposing a modular method to probabilistically integrate sets of single-hazard vulnerability models that are being constantly developed and calibrated by experts from various research fields to be used within a multi-risk context. The rest of this introduction will discuss the state of the art in exposure modelling for large-scale building portfolios for multi-hazard risk assessment, focusing on the underlying assumptions to propose generalised building typologies with associated fragility functions used to assess their physical vulnerabilities to earthquake and tsunami. Having done that, the last part of the introduction summarises the general scope and capabilities of the original method that will be described in detail afterward.'*

**5. INTRODUCTION.** At the very end (new paragraph) tell us how the rest of the paper will be organized.

>>The last two paragraphs of the Introduction have been almost entirely rephrased. The first one refers to the proposed method whilst the second one does to the application part. Now, we are clearly announcing since this earlier stage that the method is made up of four distinctive modules, while mentioning the section of the paper where their respective explanations and details are provided. We consider this has improved the overall quality of the paper (this is also related to the last comment of the Editor about making the manuscript "a bit more user friendly in its structure". The new version of these two paragraphs are presented as follows:

*This study proposes a modular method to probabilistically integrate existing sets of single-hazard vulnerability models (or "reference schemes"). For this aim, this method comprises four main modules. The first two ones refer to sets of compatibilities between the vulnerability models selected for each single-hazard vulnerability (e.g., between existing seismic and tsunami building classification schemes). The first probabilistic compatibility set are obtained between (1) building classes (as presented in Sect. 2.1), whilst the second is obtained between (2) damage states (Sect. 2.2). These two conversions are done through the use of taxonomic attributes that are independent to the definition of the reference schemes. This is done with the purpose of representing the damage distribution resulting after the first hazard (i.e., earthquake) through a damage-updated exposure model whose damage scale is dependent on the classification scheme required for assessing the vulnerability to a triggered event (i.e., tsunami). The third module results from the need to perform risk assessment for*

*the triggered hazard using the damage-updated exposure model that is now represented in terms of the second vulnerability scheme (e.g., building classes and damage states for tsunami fragility). Hence, this module comprises the proposal of (3) sets of state-dependent fragility functions for the second hazard (e.g., tsunami), as presented in Sect. 2.3. These three modules are valuable inputs for ultimately assessing the expected cumulative damage. They are later complemented by a last fourth module: (4) a consequence model to assess the incremental direct economic losses (Sect 2.4) that are expected from consecutive hazard scenarios.*

*In the application chapter of this paper (Sect. 3), we demonstrate the application of this method by investigating the likely cumulative damage on the residential buildings of Lima (Peru) by considering this city's exposure to six mega-thrust earthquake scenarios (main shock) and subsequent tsunamis. This is done using existing vulnerability models per hazard, and addressing the probabilistic compatibilities between building classes and damage states. Complementarily, a set of tsunami state-dependent fragility functions that are obtained through the use of simple ad-hoc scaling factors are proposed. Nonetheless, as it will be discussed, these functions can and should be replaced by other sets of state-dependent tsunami fragility functions derived from more sophisticated methods when they become available. Every damage distribution is translated into direct economic losses to gain a comparative risk metric and disaggregate the contribution of each hazard scenario.*

**6. GENERAL.** All variables must be clearly defined where they are first introduced. For example, IM^A, it is not clear if this is "IM" or I and M. Although it might be obvious that A is a constant, this needs to be stated. Please go through and ensure that all variabilities have been defined where they are first introduced. Please ensure that equations have references where appropriate.

>>I have ensured that all of the variables are clearly defined where they were first introduced, as well as that equations are appropriately referenced. Regarding the punctual example of $IM^A$, the next modification has been included to provide a clear understanding. Please note that 'IM' had been previously defined several lines before.

*For example, one building that is expected to be affected by a first hazard intensity measure $IM^A$ (here A refers to an IM used to model ground-shaking (e.g. PGA in g)) and a second one $IM^B$ (B refers to an IM used to model inland tsunami inundation (e.g. inundation depth in m))*

**7. Table of variables and acronyms.** Because of the number of variables, your paper will be easier to read if you have a table of variables (and I suggest a separate part of it with acronyms), introduced early on. This can include units.

>>This suggestion is the only one that has not been incorporated in the updated version of the manuscript. We consider that with the modifications on the text where all the variables used in equations, text, and figures are now clearly defined, the table of variables is no longer needed and would be repetitive. It is clear to us that, the former modifications, as well as new improvements (i.e., changes to table 1, and by including explicit explanations to the captions of every figure), can satisfactorily replace the aim of a table of variable and acronyms. Thanks for your understanding on this matter.

**8. Figure 1.** Currently black. I think it has been inverted? Ensure that colour is not the only distinguishing feature (e.g., you could take the 'red' and also make it a dashed line) as there are colour blind people.

>>The black background of this figure has been fixed in the updated version of the manuscript. Figure 1-b has been modified. First, a schematic overview of how each component will look like after its respective development in the application section is displayed. This is aligned with one of the former comments by one of the anonymous reviewers who suggested to include a link of schematic workflow at an earlier stage. Moreover, following the Executive Editor's suggestion, the modules (i.e., the four parts of the equation on Figure 1-b) are now being shown by enclosing boxes with distinctive line styles (i.e., dotted/dashed/ continuous/ double line). We agree that this modification will be helpful for colour blind people.

**9. GENERAL.** Figure caption and table headers. Make sure these are self-standing, so that one does not need to go to the text to figure these out.

>>This suggestion has been included although the updated manuscript. Thank you.

**10. REFERENCES.** Please go again through the text and ensure that ALL facts/information/ideas which build on other people/equations have appropriate in-text citations. In most places this is fine, but I was spotting a couple examples (e.g., line 268, you state the population of Peru, but no reference). Again, this is in general well done, but please do a double check.

I have double-checked the references and I do believe that it is now complete. About the mentioned reference about the Peruvian population, the following reference has been added:

*In 2022, Peru had a population of around 33 million people, with nearly 58% of this living in coastal communities (INEI, 2022).*

This citation corresponds to: *INEI: Perú: 50 años de cambios, desafíos y oportunidades poblacionales, Instituto Nacional de Estadistica e Informatica (INEI; Institute of Statistic and Informatics), 2022. Available on: https://www.inei.gob.pe/media/MenuRecursivo/publicaciones_digitales/Est/Lib1852/libro.pdf*

**11. Figure 4.** Please put the (a), (b) (c) as part of the figure, and below the sections, not above. Will read better. If you use an acronym in the figure (TIH) then you need to define this in the figure caption "Expected tsunami inundation heights (TIH) in m for three out of six...".

*Figure 1. Expected tsunami inundation heights (TIH) in meters (m) for three out of the six considered scenarios per moment magnitude (Mw), namely: (a) Mw 8.6; (b) Mw 8.8; and (c) Mw 9.0.*

**12. Table 1.** I found it difficult to read 'quickly' going from left column to middle, as acronyms are not always defined (e.g., RC, which I think means reinforced) and acronyms are only used once. Consider having another part of the table or another table which has a list A to Z of the acronyms, specific for this table? So I figured it out 'after' reading it through a couple times, but it was not user friendly (So

I needed to 'interpret' that H1 is 1 story, H3 is three stores, that MUR for line one is unreinforced masonry, etc.

>> Thanks a lot for your helpful feedback. After including your suggestions, I have payed attention that all of the building attribute values contained in the GEM v.2.0 taxonomy are appropriately and successively mentioned in the updated version of the manuscript. This means that each attribute is only reused to describe a building typology only if it was previously defined. I have asked a few external colleagues to please read again the table and check if now they can understand the building classes once each acronym was already defined, and their response was positive. I feel it is now much more organised. Thanks again for this meaningful suggestion that has improved the quality of presenting this relevant information.

**13. Table 1 and GENERAL.** Any time that monetary amounts are stated (e.g., USD/bdg) then you need to state for what year it has been normalized to.

>> After including your suggestions, the header of this table has been updated as follows:

*Table 1. SARA building classes proposed for the residential building stock of Metropolitan Lima and Callao, with their respective replacement costs per building unit (Repl. Cost (USD/bdg.) as reported in Yepes-Estrada et al., (2017) in the frame of the SARA model released by GEM (Global Earthquake Model) in 2015, which was based on official census data reported by INEI, (2007). The intensity measures (IM) of the associated seismic fragility functions to each building class, as reported in Villar-Vega et al., (2017), are also provided.*

**14. Figure 5.** Black should not be black? Cannot read any other text above legend or surrounding map, because of the black. Do you mean to have your beginning and ending points exactly the same for your ranges? For example, 23.2 is repeated as upper end of one range and lower end for another range. For this figure, please label the separate parts (a), (b), (c) and then refer to these in the figure caption (which gets around 'just' using colour, which does look nice, but not everyone will be able to see).

>> The black background of this figure has been fixed. I fully understand the Editor's concern about the fact that the ranges on this figure are repeated. The reason behind this is that our preferred manner to show these percentages is by only using one decimal value. We do not consider that the overall concept or aim of this figure will change if a second decimal value is added. The interpretation of this figure by the reader will remain the same. Thanks for your understanding on this matter. Complementary, after including your suggestions, the caption of this figure has been updated by including these lines that separate the parts (a), (b), (c) as suggested:

> *"(a) masonry and earthen (red); (b) reinforced concrete, RC and Unknown, UNK (blue); (c) wooden types (green)".*

This new version mentions the main material of the building typologies in the first place and the colour is a secondary descriptor mentioned at the end of each class, and hence. Thanks for your suggestion.

**15. Figure 6 caption.** This is an example of making it complete. The reader will have no idea what all the acronyms mean, so either define them here (I recognize this might take too much space) or refer the reader to a table where they are defined "Acronyms for building class component types are given in Table *".

>>Thank you for your kind comment. After including your suggestions, the caption of this figure has been updated by including the following lines:

> *Acronyms for SARA building classes are given in Table 1, whilst the six Medina (2019) classes are: M-PN (wooden), M-MP (masonry), M-PCP1-T1 (framed RC, one storey with similar length-width ratio), M-PCP1-T2 (framed RC, one storey, with a higher length to width ratio), M-PCP2 (framed RC, 2 storeys), and M-PCP3 (framed RC, 3 or more storeys).*

**16. Figure 7.** Is this meant to be black? In all figure captions, ensure that acronyms are defined. If someone were to remove this figure without the text, they would not know what AeDES means. Because there are only four of them, define VS, FL, RF, IP.

>> The black background of this figure has been fixed. After including your suggestions, the updated caption of this figure is:

> *Figure 7. Examples of the AeDES-based heuristics (see original AeDES form (Baggio et al., (2007) on Figure 2)) that describe the expected observable damage onto the four selected building components listed in Eq. 3 (vertical structure (VS); floor (FL); roof (RF); infills and partitions (IP)) using the scale from I-A (i.e., I=0 (null) to A=9 (>2/3 extension within the "very heavy" damage level). This is done per damage state per building class within two hazard-dependent vulnerability schemes.*

**17. Section 6.** Something happened here to your font. It is now all bold.

>> Sorry about this. The last paragraph of the discussion was merged with the format of Section 6. It has been fixed in the updated version of the manuscript.

**18.** The manuscript has some very detailed and interesting research, but would benefit overall to be made a bit more user friendly in its structure, making it easier for the reader to go from one section to another 'quickly' without having to read in detail (a couple of times) a given section. I've given some notes above of some items that might be changed, but you might want to consider how you can really make the paper useable to the average reader.

>> Indeed, after having included the Executive Editor's suggestions, the quality of the paper has been largely improved. Several new changes in the format have made it possible. We would like to express our deepest gratitude for the time invested on this matter.

---

## Author Response (AR3)

Dear NHESS Editorial office,

Thank you for your positive response and for having accepted our manuscript for publication.

The technical correction made by the publishing office about the caption of Figure 11 that contains maps and aerials has been solved.

With best regards,

Juan Camilo Gomez Zapata